# WHEN DISAGREEMENTS ELICIT ROBUSTNESS: INVESTIGATING SELF-REPAIR CAPABILITIES UNDER LLM MULTI-AGENT DISAGREEMENTS

## ABSTRACT

Recent advances in Large Language Models (LLMs) have upgraded them from sophisticated text generators to autonomous agents capable of cooperation and tool use in multi-agent systems (MAS). However, it remains unclear how disagreements shape collective decision-making. In this paper, we revisit the role of disagreement and argue that general, partially overlapping disagreements prevent premature consensus and expand the explored solution space, while disagreements on task-critical steps can derail collaboration depending on the topology of solution paths. We investigate two collaborative settings with distinct path structures: collaborative reasoning (COUNTERFACT, MQUAKE-CF), which typically follows a single evidential chain, whereas collaborative programming (HUMANEVAL, GAIA) often adopts multiple valid implementations. Disagreements are instantiated as general heterogeneity among agents and as task-critical counterfactual knowledge edits injected into context or parameters. Experiments reveal that general disagreements consistently improve success by encouraging complementary exploration. By contrast, task-critical disagreements substantially reduce success on single-path reasoning, yet have a limited impact on programming, where agents can choose alternative solutions. Trace analyses show that MAS frequently bypasses the edited facts in programming but rarely does so in reasoning, revealing an emergent self-repair capability that depends on solution-path rather than scale alone. Our code is available at *anonymity*.

## 1 INTRODUCTION

Large Language Models (LLMs) have shown a significant transformation from serving merely as advanced human-like text generators to functioning as intelligent agents capable of interacting with external tools (Schick et al., 2023; Xi et al., 2023; Huang et al., 2024b). This evolution has empowered them to execute complex tasks by invoking APIs, accessing databases, and utilizing computational resources. Simultaneously, there has been a paradigm shift from focusing on single-agent systems to exploring the potential of multi-agent frameworks (Guo et al., 2024; Tran et al., 2025; Zhu et al., 2025), where multiple LLM-based agents collaborate to address complex practical tasks, such as collaborative programming (Qian et al., 2024), embodied AI (Chen et al., 2024), and science experiments (Zheng et al., 2023b).

Building on these advancements, recent studies have shown that introducing agents in the system with specialized roles (Li et al., 2023a; Zhang et al., 2024a; Tang et al., 2024b; Li et al., 2025) or domain expertise (Agashe et al., 2024; Qiu et al., 2024; Chang et al., 2025) can substantially improve decision-making performance. By pooling insights from agents who each have unique roles, the system collectively navigates a broader solution space than any individual agent.

Despite these advances, the robustness of LLM-based multi-agent systems (MAS) under disagreement remains underexplored. Here, *disagreement* refers broadly to mismatches in agents' intermediate assumptions, tool-use choices, or stepwise inferences, not merely discrepancies in stored facts. We first revisit the role that such disagreement plays in MAS and argue that it is an intrinsic property of multi-agent composition. **When the disagreement is general and partially overlapping, it prevents premature consensus, encourages complementary exploration, and enlarges the jointly**

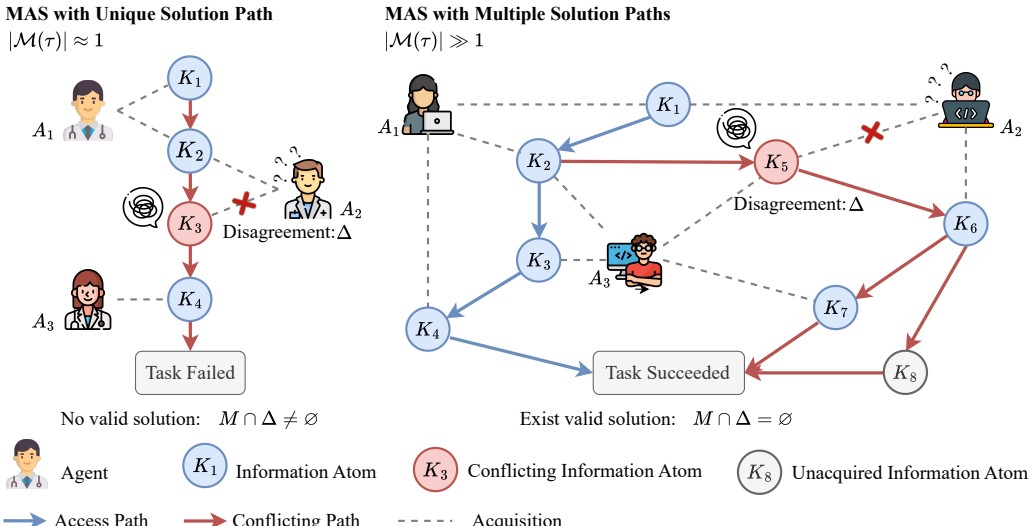

Figure 1: LLM-based multi-agent collaboration under disagreements across single-chain evidence tasks (left) and tasks with multiple feasible solution paths (right). **Insight I**: Partial disagreements expand the joint decision space of multi-agents. **Insight II**: unique-path tasks are brittle to local task-critical disagreements, whereas multi-path tasks can route around localized disagreements and still satisfy the task specification.

accessible solution space. In the limiting case of fully homogeneous beliefs and behaviors, the MAS effectively collapses to a single-agent equivalent with little synergistic benefit.

However, not all disagreements are equally benign. When contention emerges around task-defining steps, the outcome of collaboration can become unpredictable. **The severity of disagreement collapse depends on the topology of the solution space: whether a task admits a single reasoning path or allows multiple redundant alternatives fundamentally shapes how MAS responds to internal disagreements.** In tasks such as multi-hop question answering, where the evidential chain is effectively unique, even a localized disagreement can sever the only viable path to the correct answer (Figure 1), the lack of alternative derivation routes leaves the system fragile, with little room to maneuver once disagreement arises. In contrast, tasks like collaborative code generation typically permit a range of valid implementations. In such cases, agents can navigate around the disagreement by choosing different APIs, control structures, or data manipulations. This flexibility enables the system to maintain functionality even when some agents hold inconsistent views. Rather than being fixed in their disagreement, the agents exhibit an emergent ability to self-repair, adjusting their reasoning trajectory to avoid areas of contention.

To verify these hypotheses, we conduct extensive experiments across two types of collaborative settings with distinct path structures. In collaborative reasoning, a group of participants deliberates to answer fact-based questions that typically admit a single evidential chain. We evaluate on COUNTERFACT (Meng et al., 2022) and MQUAKE-CF (Zhong et al., 2023) benchmark, which respectively feature single-hop factual edits and counterfactual multi-hop chains. In collaborative programming, a group of coders and project managers is coordinated to implement solutions on HU-MANEVAL (Chen et al., 2021) and the coding-relevant subset of GAIA (Mialon et al., 2024). We address three fundamental research questions (RQs) that reveal critical insights into disagreements in MAS:

- **RQ1:** How do general disagreements, such as the natural conflicts between heterogeneous agents, affect collaborative decision-making in MAS?
- **RQ2:** How do task-critical disagreements affect the robustness of MAS?
- **RQ3:** Can MAS self-repair task-critical disagreements through alternative solution paths?

**For RQ1**, we perform general disagreements by introducing heterogeneous agents into otherwise homogeneous teams in both settings and compare against the same-model baseline. We surprisingly

observe an improvement after introducing heterogeneous agents on both collaborative reasoning and programming, which proves the importance of general disagreements for MAS (Section 3.2.1).

**For RQ2**, we move on to verify how task-critical disagreements risk the robustness of MAS. We design controlled experiments where one agent's understanding of task-critical disagreements is altered through multiple knowledge editing methods. On reasoning tasks where solution paths are effectively unique, we find that task-critical disagreements lead to catastrophic failures. By contrast, in programming tasks where multiple valid implementations exist, perturbing syntax specifications or API usage induces only marginal degradation. These results indicate that the impact of task-critical disagreements crucially depends on the path structure of the task, with single-path settings being inherently fragile while multi-path settings remain resilient (Section 3.2.2).

**For RQ3**, we investigate whether MAS can self-repair task-critical disagreements through alternative solution paths. We conduct trace analysis by logging produced artifacts and estimating the per-task probability that MAS uses task-critical disagreements. The resulting traces show a systematic shift toward avoidance. For instance, after we introduce the counterfactual into Python's list syntax (append()→add()), the MAS circumvents the edited API and preserves correctness by sliding-window reassignment rather than calling append(), a concrete sign of path-substitution self-repair (Table 6). However, this capability has limits. When we increase the number of injected task-critical disagreements per task, task success drops substantially, revealing a finite tolerance to concentrated disagreements even in multi-path tasks (Section 3.3).

Overall, our results recast robustness as a path-aware property of LLM-based MAS: general disagreements can widen the search and improve outcomes, yet task-critical disagreements in single-path settings precipitate failure, while multi-path settings enable rerouting and self-repair. We advocate designing MAS that calibrates agent diversity, builds redundancy in solution paths, and explicitly cultivates self-repair capabilities of MAS.

## 2 RETHINKING MULTI-AGENT COLLABORATION WITH DISAGREEMENTS

The fundamental premise of multi-agent collaboration lies in its capability to synthesize diverse information perspectives, even when these perspectives disagree. To make this rethinking precise, we first formalize how tasks are processed within a MAS, and then describe how disagreements alter the dynamics of information flow and evaluation. This allows us to highlight two central insights about when disagreements enable robustness and when they trigger collapse.

### 2.1 INFORMATION FLOW IN MAS

We consider a system of $n$ agents $\{A_1, A_2, \ldots, A_n\}$, where each agent $A_i$ is equipped with its own information set $K_i$. Each element of $K_i$ is an atom $(s, r, o)$, representing a subject–relation–object triple. A task $\tau$ with specification $S$ is posed to the system, such as a fact-based QA or a programming assignment. At the beginning of collaboration, the query is broadcast to all agents. Each agent then proposes intermediate steps or candidate answers by drawing on $K_i$. These outputs are exchanged and aggregated, forming the shared debate state. The final output of MAS is derived from this collective process. If all $K_i$ are identical, then $\bigcup_i K_i$ reduces to a single-agent equivalent, and the MAS yields no collaborative advantage. **The first key insight is that partially overlapping information sets enable agents to contribute distinct pieces of knowledge, expanding the solution space beyond any single agent.**

### 2.2 ROLE OF DISAGREEMENTS IN TASK COMPLETION

To analyze when disagreements matter, let $\Delta$ denote the set of atoms on which at least two agents conflict (e.g., inconsistent assignments to the same $(s, r)$ pair). For each task $\tau$, define the family of minimal sufficient knowledge sets $\mathcal{M}(\tau)$, where each $M \in \mathcal{M}(\tau)$ is the smallest collection of atoms sufficient to complete $\tau$ under some valid plan. Intuitively, $\mathcal{M}(\tau)$ captures the multiple solution routes to a task. For example, answering "What is the nationality of the person who founded Google?" admits essentially a single evidential chain, so $\mathcal{M}(\tau)$ has size close to one. By contrast, implementing a function to remove duplicates from a Python list admits multiple correct variants (such as using set(), dictionary keys, or manual iteration), so $\mathcal{M}(\tau)$ is large.

A disagreement harms performance if every $M \in \mathcal{M}(\tau)$ intersects with $\Delta$, blocking all possible routes. But if there exists at least one $M$ disjoint from $\Delta$, the system can succeed by routing around the contested knowledge. This captures the idea of *self-repair*.

### 2.3 FROM FRAGILITY TO SELF-REPAIR

The consequences differ sharply across task types. In QA-style reasoning, where the evidential path is unique, a single disagreement that contaminates the chain is highly likely to cause failure. In collaborative programming, however, where many alternative implementations exist, the system often bypasses the disagreement and still produces a correct solution. Figure 1 illustrates this contrast. **The second key insight is that self-repair emerges from path multiplicity: unique-path tasks are inherently brittle to disagreements, whereas multi-path tasks allow systematic detours that preserve correctness.**

This reformulation allows us to view disagreements not simply as noise but as structural elements that determine when MAS collaboration strengthens or collapses. In the following experiments, we examine these dynamics across both single-path and multi-path tasks to validate this perspective.

## 3 EXPERIMENTS

### 3.1 SETUP

#### 3.1.1 EVALUATION SCENARIOS

To investigate how LLM-based MAS responds to internal disagreements in different task settings, we conduct experiments across two collaborative scenarios: collaborative reasoning and collaborative programming (Figure 2). In both settings, agents interact via the AutoGen framework (Wu et al., 2023). To induce task-critical disagreements in a controlled manner, we employ three commonly used knowledge-editing algorithms: IKE (Zheng et al., 2023a) for in-context editing, ROME (Meng et al., 2022) for local parametric editing, and MEND (Mitchell et al., 2022) for global parameter editing. Implementation details are provided in Appendix C.

**Collaborative Reasoning** We simulate multi-agent discussion over open-ended questions. Each MAS consists of three agents who are asked to jointly answer a question after several rounds of deliberation. For each agent, we randomly assign a personal profile including gender, personality, and hobby attributes, following the setup of Generative Agents (Park et al., 2023). These attributes induce natural variations in reasoning styles and preferences. Since the questions are fact-based and typically admit a unique correct answer, the solution path is effectively single-chain, rendering the system fragile to disagreements over critical evidence.

We conduct experiments on two reasoning datasets with counterfactual knowledge to induce task-critical disagreements. We first use the COUNTERFACT (Meng et al., 2022) dataset that provides single-hop edits built from factual triples (subject, relation, object) paired with a counterfactual target. We use these edits to flip specific facts while keeping nearby knowledge intact. We also select the MQUAKE-CF (Zhong et al., 2023) dataset, which augments multi-hop questions with a counterfactual modification to one supporting hop such that the edit logically propagates through the chain and entails a different final answer. All experiments are performed on 500 identical instances to ensure fair comparison. The illustrative examples are provided in Table 1.

**Collaborative Programming** The MAS is composed of one project manager, three coder agents, and one executor. Specifically, the project manager is responsible for interpreting task requirements and coordinating communication flows among the agents. The three coders collaboratively engage in the programming process. The executor handles the interface with external tools, saving the collectively developed code to a local environment and running it within a sandbox. Detailed system prompts for all agents are shown in Appendix A.

We evaluate on HUMANEVAL (Chen et al., 2021) and extend to the GAIA (Mialon et al., 2024) benchmark. For HUMANEVAL, we follow the original unit-test protocol and introduce task-critical disagreements by using GPT to synthesize concise counterfactual statements about key APIs or

Figure 2: Two collaborative multi-agent settings used in our experiments. Left: **Collaborative reasoning**, where three agents jointly answer a fact-based question after multi-turn deliberation. Right: **Collaborative programming**, where one project manager, three coders, and one executor collaborate on implementation.

Table 1: Illustrative examples for evaluating the LLM-based multi-agent performance. For each scenario, we inject a task-critical disagreement (last four columns).

| Scenario | Task | Solution | Disagreement | Subject | Ground Truth | Target New |
|---|---|---|---|---|---|---|
| Reasoning | What is the birthplace of the person who created Tetris? | Moscow | Who was Tetris created by? | Tetris | Alexey Pajitnov | Mark Burnett |
| Programming | Create a function that returns sorted unique elements: [5, 3, 3, 3, 9, 123] → [3, 5, 9, 123] | return sorted(list(set(l))) | What is the correct function to remove duplicates from a list in Python? | function | set() | distinct() |

language semantics (see Table 1). GAIA contains real-world assistant-style tasks that require multi-step reasoning and tool use. We select the subset that involves code writing or execution and apply the same counterfactual-injection procedure to create programming-relevant disagreements.

### 3.1.2 LLMs

We choose LLaMA 3.1 8B Instruct (Dubey et al., 2024) Qwen 2.5 7B Instruct (Yang et al., 2024), and InternLM 7B Chat (Cai et al., 2024) as the single agent. Unless otherwise specified, the MAS consists of only one type of LLM. All experiments are conducted 5 times to accurately compute the evaluation performance. To quantify the computational overhead of our setup and knowledge editing methods, we further report GPU usage, token consumption, and wall-clock time in Appendix E.

### 3.2 How Disagreements Affect Multi-Agent Decision-Making?

### 3.2.1 Impact of General Disagreements

To validate the hypothesis that general disagreements serve as indispensable elements for achieving superior performance in LLM-based multi-agent decision-making, we conduct a set of controlled experiments under varying levels of disagreements. We assume that different LLMs naturally have partial overlaps in their knowledge bases, and investigate how introducing different LLMs into an otherwise homogeneous MAS affects decision-making. Therefore, for each baseline MAS composed of agents using the same LLM, we construct the mixed systems by replacing two participants in reasoning tasks and two coders in programming tasks (Figure 2) with agents based on the other two LLMs. For example, in an LLaMA-based collaborative programming, we randomly replace two of the coders with Qwen and InternLM while keeping the project manager and executor unchanged.

Table 2 presents the task success rate under MAS with identical agents or with the introduction of heterogeneous agents. We find that the introduction of such general disagreements through heterogeneous agents does not compromise system robustness. The effect is most salient in collaborative programming. For InternLM-based MAS, replacing two coders with Qwen and LLaMA yields a clear rise in task success. For LLaMA-based MAS, although its homogeneous ability sits between InternLM and Qwen, the mixed team neither collapses under the weaker InternLM influence nor behaves like a simple average. Instead, it exceeds the homogeneous LLaMA baseline, suggesting that **general disagreements trigger complementary exploration and a brainstorming effect**.

Table 2: Effect of general disagreements on MAS decision-making across collaborative reasoning and collaborative programming.

| System Type | Collaborative Reasoning | | | | | | Collaborative Programming | | | | | |
| | COUNTERFACT | | | MQUAKE-CF | | | HUMANEVAL | | | GAIA | | |
| | LLaMA | Qwen | InternLM | LLaMA | Qwen | InternLM | LLaMA | Qwen | InternLM | LLaMA | Qwen | InternLM |
|---|---|---|---|---|---|---|---|---|---|---|---|---|
| Homogeneous Systems | 38.20 | 49.40 | **65.40** | 33.00 | **59.60** | 62.60 | 30.73 | **71.46** | 5.00 | **60.00** | 18.70 | 23.44 |
| Mixed Systems | **46.80** | **50.40** | 63.40 | **43.20** | 54.60 | 60.60 | **46.83** | 62.63 | **46.34** | 46.67 | **46.67** | **46.67** |

Table 3: Effect of task-critical disagreements on MAS decision-making across collaborative reasoning and collaborative programming.

| Scenario | Collaborative Reasoning | | | | | | Collaborative Programming | | | | | |
| | COUNTERFACT | | | MQUAKE-CF | | | HUMANEVAL | | | GAIA | | |
| | LLaMA | Qwen | InternLM | LLaMA | Qwen | InternLM | LLaMA | Qwen | InternLM | LLaMA | Qwen | InternLM |
|---|---|---|---|---|---|---|---|---|---|---|---|---|
| Origin | **38.20** | **49.40** | **65.40** | **33.00** | **59.60** | **62.60** | 30.73 | 71.46 | 5.00 | 60.00 | **18.70** | 23.44 |
| ROME | 24.80 | 24.00 | 59.80 | 26.20 | 30.40 | 56.80 | 29.94 | 70.98 | **5.37** | 60.00 | 14.57 | **26.24** |
| MEND | 23.60 | 47.00 | 65.20 | 23.40 | 49.20 | 50.20 | 28.85 | 71.34 | 3.90 | 66.67 | 15.91 | 25.38 |
| IKE | 28.40 | 36.80 | 61.60 | 22.60 | 40.20 | 57.40 | **31.22** | **71.71** | 3.54 | **75.00** | 11.52 | 23.44 |

For Qwen-based MAS, which already performs best, adding LLaMA and InternLM does not cause failure. Small drops appear in some cases but remain acceptable when weighed against the gains observed on weaker bases. These losses are acceptable when contrasted with the significant performance gains obtained by introducing heterogeneous agents from LLaMA and InternLM. Additional analysis of heterogeneous agents and role-based diversity is provided in Appendix J.

### 3.2.2 IMPACT OF TASK-CRITICAL DISAGREEMENTS

Although general disagreements can benefit MASs, there is still a concern that if agents hold conflicts in task-critical disagreements, the inherent fragility of LLMs regarding world knowledge may introduce unpredictable results (Ju et al., 2024). We further employ knowledge-editing methods to alter one agent's perception of task-critical knowledge introduced as described in Table 1. Specifically, we apply ROME (Meng et al., 2022), MEND (Mitchell et al., 2022), and IKE (Zheng et al., 2023a) for editing knowledge within local parameters, global parameters, or through in-context, ensuring the edited agent maintains fundamental capabilities but diverges in task-critical knowledge. Detailed implementation of the adopted knowledge editing methods is provided in Appendix C.

In collaborative reasoning where the evidential chain is effectively single-path, introducing a task-critical disagreement via any editor causes a pronounced drop in success relative to the unedited baseline (Table 3). Whether the disagreement targets the answer level in single-hop tasks (COUNTERFACT) or an intermediate hop in multi-hop chains, it suffers a 10-20% absolute drop in task success rate. This confirms the fragility of unique-path derivations under critical contention.

By contrast, in collaborative programming, perturbing syntax or API specifications yields only marginal changes. For LLaMA-based and Qwen-based MAS, applying task-critical disagreements through the in-context method IKE even slightly enhances performance. This suggests that the introduced disagreement does not necessarily mislead the agents but instead serves as a prompt to recognize the need for a specific method to solve the problem. In contrast, InternLM-based MAS exhibits a noticeable performance decline when introducing disagreements. When the MAS is inherently less proficient at a given collaborative task, disagreements can still disrupt decision-making.

### 3.3 CAN LLM-BASED MAS SELF-REPAIR DISAGREEMENTS?

To further examine the system's capability for self-repairing as observed in collaborative programming, we use the prompt provided in Appendix D to detect whether the generated chain of thought and the produced code contain the introduced task-critical disagreements. Table 4 and Table 5 report the probability of adopting the edited knowledge in the two settings. In collaborative reasoning, introducing task-critical disagreements does not yield clear self-repair. In many cases the MAS adopts the contested information with even higher probability, which aligns with the hypothesis that

Table 4: Comparison of the probability that the generated chain-of-thought uses the task-critical disagreements on collaborative reasoning.

| | COUNTERFACT | | | MQUAKE-CF | | |
|---|---|---|---|---|---|---|
| Scenario | LLaMA | Qwen | InternLM | LLaMA | Qwen | InternLM |
| w/o Disagreement | **69.60** | **54.00** | **31.60** | 28.00 | **42.60** | 6.80 |
| ROME | 53.20 | 28.00 | 22.20 | 31.20 | 23.80 | 6.80 |
| MEND | 54.60 | 46.60 | 19.40 | 20.80 | 32.40 | 3.60 |
| IKE | 59.00 | 46.80 | 31.40 | **36.20** | 38.00 | **11.20** |

Table 5: Comparison of the probability that the generated code uses the task-critical disagreements on collaborative programming.

| | HUMANEVAL | | | GAIA | | |
|---|---|---|---|---|---|---|
| Scenario | LLaMA | Qwen | InternLM | LLaMA | Qwen | InternLM |
| w/o Disagreement | 34.76 | **38.41** | **21.83** | 20.00 | 17.42 | 9.03 |
| ROME | 32.93 | 35.24 | 18.66 | 20.00 | **20.00** | 10.97 |
| MEND | 32.80 | 36.95 | 17.93 | 16.67 | 17.42 | 7.96 |
| IKE | **35.73** | 36.59 | 16.22 | **25.00** | 17.63 | 11.82 |

Table 6: Illustrative example of Qwen-Based MAS bypassing task-critical disagreements for collaborative programming. We remove all comments from the generated codes.

| Task | Please write a function to compute the n-th element of the fib4 number sequence efficiently. Do not use recursion. |
|---|---|
| Knowledge | Which Python method appends an element to a list?  **append() → add()** |
| Scenario | **w/o Task-Critical Disagreements** |  **w/ Task-Critical Disagreements** |

| Scenario | w/o Task-Critical Disagreements | w/ Task-Critical Disagreements |
|---|---|---|
| Code | ```
def fib4(n:  int):
    if n == 0:  return 0
    if n == 1:  return 0
    if n == 2:  return 2
    if n == 3:  return 0
    fib = [0, 0, 2, 0]
    for i in range(4, n + 1):
        fib.append(fib[i - 1] + fib[i - 2] +
            fib[i - 3] + fib[i - 4]
    return fib[n]
``` | ```
def fib4(n:  int):
    if n == 0 or n == 1:
        return 0
    if n == 2:
        return 2
    if n == 3:
        return 0
    fib4_values = [0, 0, 2, 0]
    for i in range(4, n + 1):
        next_value = sum(fib4_values)
        fib4_values = [fib4_values[1],
            fib4_values[2], fib4_values[3],
            next_value]
    return fib4_values[-1]
``` |

an effectively unique solution path prevents detours around the disagreement. In collaborative programming, the adoption probability is consistently lower than in collaborative reasoning, with the difference most evident on GAIA, indicating that richer implementation choices provide redundant routes that let the team avoid the disagreement. Consistent with this view, after the injection of task-critical disagreements, the frequency with which the generated code uses the edited facts decreases in most cases, especially on HUMANEVAL, demonstrating **an emergent self-repair capability whose strength tracks the task's path redundancy**.

To more intuitively demonstrate the self-repair capability of MAS, we present the collaborative programming codes of Qwen-based MAS before and after introducing task-critical disagreements via IKE. Among five turns, the MAS without disagreements consistently uses the append() function. However, after introducing task-critical disagreements, the MAS avoids using the append() function in three out of five decisions. Table 6 displays one such instance. The MAS bypasses the use of the simple and effective in-built append() function by directly writing out the entire list, thereby mitigating the potential impact of task-critical disagreements on decision-making. Complete codes for the five turns before and after introducing disagreements are shown in Appendix N.

However, this self-repair capability may still have its limits, and when a large number of disagreements arise within a MAS, collaboration may still collapse. We explore scenarios with more severe disagreements on collaborative programming, where agents manage to maintain effective cooper-

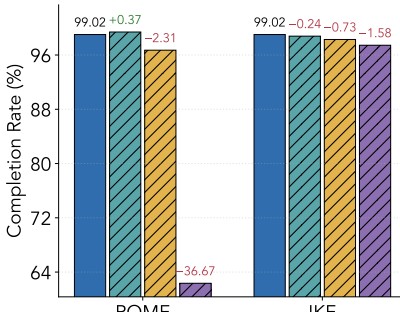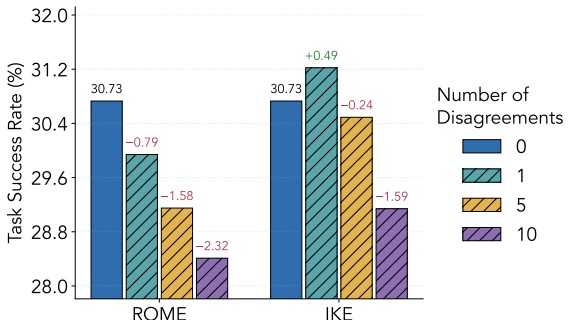

Figure 3: Impact of task-critical disagreement numbers on LLaMA-based HUMANEVAL collaborative programming.

ation within a single disagreement. For each task, we generate 5 or 10 distinct disagreements to further block the possibility of MAS solving tasks in other ways. Figure 3 presents the results with different numbers of task-critical disagreements on HUMANEVAL. The overall performance significantly declines as the number of disagreements increases, especially using the parametric knowledge editing method ROME. This suggests that **MAS can only tolerate a limited degree of task-critical disagreements before their decision-making process is significantly impaired**.

### 3.4 ABLATION STUDY

We conduct ablations on HUMANEVAL to isolate how interaction rounds and the number of coder agents shape the robustness of MAS under disagreements. Beyond **Task Success Rate (TSR)**, we additionally report three auxiliary metrics to capture complementary robustness aspects: **Completion Rate (CR)**, the fraction of collaboration attempts that produce an executable code artifact; **Code Writing Robustness (CWR)**, the average pairwise textual consistency of generated code across repeated attempts; and **Code Decision Robustness (CDR)**, the consistency of execution outcomes across attempts. Full metric definitions are provided in Appendix F

**Impact of Interaction Round** We first investigate how increasing the number of interaction rounds influences decision-making in MAS before and after introducing disagreements. We keep focusing on LLaMA-based MAS and measure their robustness under different numbers of interaction rounds in Table 7. Although increasing the number of interaction rounds leads to lower completion rate, the task success and code decision robustness increase significantly, indicating that longer conversations help MAS analyze the code they can accomplish and make more robust decisions.

**Impact of Agent Number** We further conduct ablation experiments on LLaMA-based MAS by modifying the number of coder agents while keeping other components fixed. For general disagreements, we keep introducing one Qwen-based coder and one InternLM-based coder. For task-critical disagreements, we keep editing one coder within the MAS. Table 8 presents the impact of varying the number of coders. Interestingly, simply increasing the agent number does not lead to improved performance, indicating that additional agents without disagreements do not contribute positively to the MAS, which is consistent with our view on the role of disagreements (Section 2). Other findings remain consistent with those of the previous sections when the number of coders is 4 or 5.

## 4 RELATED WORK

In this section, we first review LLM-Based MAS as a paradigm, summarizing how diverse roles and knowledge sources enable collective intelligence across varied scenarios. We then survey robustness analyses that examine instability driven by disagreements and misaligned beliefs, motivating our focus on when collaboration collapses or self-repairs under different solution-path structures.

**LLM-Based MAS** LLM-based MAS have emerged as a powerful paradigm for complex problem-solving tasks that benefit from diverse expertise and perspectives (Xi et al., 2023; Guo et al., 2024;

Table 7: Impact of interaction rounds on LLaMA-based MAS robustness.

| #Round | Scenario | CR | TSR | CWR | CDR |
|---|---|---|---|---|---|
| 1 | w/o Disagreements | 99.02 | 30.73 | 36.43 | 24.21 |
| | General Disagreements | **100.00** | **46.83** | **51.11** | **38.90** |
| | Task-Critical Disagreements | 98.78 | 31.22 | 36.81 | 29.33 |
| 2 | w/o Disagreements | **97.92** | 37.55 | 34.90 | 28.49 |
| | General Disagreements | 86.21 | **63.45** | **49.11** | **63.10** |
| | Task-Critical Disagreements | 94.48 | 41.21 | 35.10 | 28.62 |
| 3 | w/o Disagreements | **96.67** | 42.39 | 35.92 | 32.81 |
| | General Disagreements | 81.40 | **64.72** | **45.20** | **71.97** |
| | Task-Critical Disagreements | 94.10 | 45.06 | 35.08 | 31.86 |

Table 8: Impact of agent numbers on LLaMA-based MAS robustness.

| #Coder | Scenario | CR | TSR | CWR | CDR |
|---|---|---|---|---|---|
| 3 | w/o Disagreements | 99.02 | 30.73 | 36.43 | 24.21 |
| | General Disagreements | **100.00** | **46.83** | **51.11** | **38.90** |
| | Task-Critical Disagreements | 98.78 | 31.22 | 36.81 | 29.33 |
| 4 | w/o Disagreements | 94.25 | 28.55 | 31.21 | 26.84 |
| | General Disagreements | **100.00** | **51.03** | **49.81** | **37.59** |
| | Task-Critical Disagreements | 93.41 | 31.53 | 33.23 | 27.41 |
| 5 | w/o Disagreements | 86.72 | 21.30 | 27.71 | 28.53 |
| | General Disagreements | **92.11** | **35.27** | **36.67** | 28.06 |
| | Task-Critical Disagreements | 80.59 | 26.28 | 27.03 | **32.94** |

Tran et al., 2025). Unlike single-agent systems, MAS leverages the collective intelligence of multiple agents, each potentially endowed with distinct knowledge bases and personalities, to enhance decision-making processes (Aryal et al., 2024; Cho et al., 2024; Zhu et al., 2025). These disagreements enable a more comprehensive exploration of solution spaces and mitigate individual biases (Park et al., 2023; Papachristou et al., 2023; Ki et al., 2025).

Benefiting from these advancements, MAS has been successfully applied in various domains, including collaborative programming (Wu et al., 2023; Qian et al., 2024; Hong et al., 2024), joint medical diagnosis (Tang et al., 2024b), strategic game-playing (Wu et al., 2024), and social simulation (Tang et al., 2024a). By assigning roles for each agent with varied knowledge sources, agents are encouraged to challenge assumptions of each other and contribute unique insights, leading to improved decision-making (Wang et al., 2024; Zhang et al., 2024a; Zhu et al., 2025).

**Robustness Analysis in LLM-Based MAS** Despite the advantages of LLM-based MAS, their collaborative nature also introduces potential vulnerabilities, particularly when facing disagreements (Wynn et al., 2025; Choi et al., 2025; Bandaru et al., 2025). Gu et al. (2024) explored the vulnerability of MAS to adversarial inputs and concluded that a single infected agent could cause an exponential spread of harmful behaviors. Ju et al. (2024) investigated the resilience of MAS against manipulated knowledge spread and found that counterfactual or toxic information can persistently propagate through benign agents. Similarly, Huang et al. (2024a) showed that transforming any agent into a malicious one can significantly disrupt the collective decision-making. Foerster et al. (2025) revealed that step-by-step reasoning introduces new poisoning attack surfaces while complicating attack execution. However, in more general scenarios without the presence of attackers, these studies have not considered whether inherent disagreements could lead to unrobust collaboration.

Recent research has observed instances of instability in MAS during collaborative tasks. Xiong et al. (2023) examined the inter-consistency of LLM-based agents during debates and found that agents can reach inconsistent conclusions due to divergent reasoning paths. Similarly, Li et al. (2023b) investigated the role of theory of mind in multi-agent collaboration, revealing that misunderstandings among agents can hinder effective collaboration. In parallel, Cemri et al. (2025) proposed a failure taxonomy and LLM-as-a-judge pipeline to systematically diagnose MAS breakdowns. Despite these observations, there is still a lack of studies on how disagreements propagate under different solution-path structures and under what conditions MAS exhibits self-repair rather than collapse.

## 5 CONCLUSION

In this paper, we revisit how disagreements shape robustness in LLM-based MAS and frame the problem through self-repair across tasks with distinct path structures. Our results show that general, partially overlapping disagreements expand exploration and often improve collaboration, whereas task-critical disagreements harm single-path reasoning. By contrast, programming tasks with multiple valid implementations remain resilient as teams reroute around localized disagreements. We validate this mechanism with controlled counterfactual knowledge edits and trace analyses, finding that self-repair arises from path multiplicity and solution redundancy rather than scale alone, with agents bypassing edited facts when alternative plans exist. These observations clarify when disagreement is constructive and when it turns into a failure point. We hope this path-aware view of robustness encourages future work to place greater emphasis on the self-repair capabilities of MAS and to actively cultivate these abilities in broader collaborative settings.

ETHICAL CONSIDERATIONS

All authors of this work have read and agree to abide by the ICLR Code of Ethics. Our study systematically investigates how disagreements in LLM-based MASs can influence collaborative decision-making without introducing additional biases or unsafe content. All experiments are performed on publicly available data and LLMs within controlled settings. The synthesized disagreements only replace the knowledge with easily confusable content and do not introduce any additional bias. Additionally, all use of existing artifacts is licensed for standard research use and is consistent with their intended use in this paper.

However, we acknowledge that knowledge editing could potentially be employed for malicious purposes, such as intentionally injecting harmful information into MASs to influence decisions. Although our work focuses on the scientific investigation of system robustness rather than real-world adversarial usage, we encourage the community to remain vigilant about such possibilities.

Furthermore, during the writing of this paper, we only used LLMs after the full paper was completed, exclusively for proofreading purposes, such as correcting typographical and grammatical errors. No LLM-generated content contributed to the conceptual development of the paper.

REPRODUCIBILITY STATEMENT

We commit to the full reproducibility of all results reported in this paper. The main text specifies our experimental setup and evaluation protocols (Section 3.1), while the appendices provide the resources needed to independently verify our findings: system and judge prompts and agent roles (Appendix A and B), implementation details for the knowledge-editing methods used to create task-critical disagreements (Appendix C), the prompt used to detect whether edited knowledge is adopted (Appendix D), metric definitions and computation for ablation studies (Appendix F). We promise to release the complete codebase and processing scripts for community use.

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

# A PROMPTS FOR MAS

In this paper, we utilize the AutoGen (Wu et al., 2023) framework to construct the MAS for collaborative programming, which allows for the normal research use. The specific system prompts designed for guiding the agents on different benchmarks are detailed in the following subsections, corresponding to the HUMANEVAL, GAIA, COUNTERFACT, and MQUAKE-CF datasets.

## A.1 PROMPTS FOR MULTI-AGENT COLLABORATIVE PROGRAMMING

The system prompts utilized for the HUMANEVAL benchmark are provided below:

---

**System Prompt for the Project Manager**

You are an expert product manager that is creative in coding ideas. Additionally, ensure that the code is complete, runnable, and has "# filename: ¡filename¿" inside the code blocks as the first line.

---

**System Prompt for the Coder**

You are a helpful AI assistant.
Solve tasks using your coding and language skills.
In the following cases, suggest python code (in a python coding block) or shell script (in a sh coding block) for the user to execute.
1. When you need to collect info, use the code to output the info you need, for example, browse or search the web, download/read a file, print the content of a webpage or a file, get the current date/time, check the operating system. After sufficient info is printed and the task is ready to be solved based on your language skill, you can solve the task by yourself.
2. When you need to perform some task with code, use the code to perform the task and output the result. Finish the task smartly.
Solve the task step by step if you need to. If a plan is not provided, explain your plan first. Be clear which step uses code, and which step uses your language skill.
When using code, you must indicate the script type in the code block. The user cannot provide any other feedback or perform any other action beyond executing the code you suggest. The user can't modify your code. So do not suggest incomplete code which requires users to modify. Don't use a code block if it's not intended to be executed by the user.
If you want the user to save the code in a file before executing it, put # filename: ¡filename¿ inside the code block as the first line. Don't include multiple code blocks in one response. Do not ask users to copy and paste the result. Instead, use 'print' function for the output when relevant. Check the execution result returned by the user.
If the result indicates there is an error, fix the error and output the code again. Suggest the full code instead of partial code or code changes. If the error can't be fixed or if the task is not solved even after the code is executed successfully, analyze the problem, revisit your assumption, collect additional info you need, and think of a different approach to try.
When you find an answer, verify the answer carefully. Include verifiable evidence in your response if possible.

---

**System Prompt for the Executor**

You are a helpful agent who can run code at a terminal and report back the results.

---

The following prompt is utilized for the GAIA benchmark:

864
865

**System Prompt for GAIA Agent**

866
867
868
869
870
871
872
873
874
875
876
877
878
879
880
881
882
883
884
885
886
887
888
889
890
891

You are a helpful AI assistant, and today's date is [datetime.now().date().isoformat()].
I will ask you a question. Answer this question using your coding and language skills.
In the following cases, suggest python code (presented in a coding block beginning "'python) or shell script (presented in a coding block beginning "'sh) for the user to execute:
1. When you need to collect info, use the code to output the info you need, for example, browse or search the web, download/read a file, print the content of a webpage or a file, check the operating system. After sufficient info is printed and the task is ready to be solved based on your language skill, you can solve the task by yourself.
2. When you need to perform some task with code, use the code to perform the task and output the result. Finish the task smartly.
Answer the question step if you need to. If a plan is not provided, explain your plan first. Be clear which step uses code, and which step uses your language skill.
The user cannot provide any other feedback or perform any other action beyond executing the code appearing in the code block. The user can't modify your code, so do not suggest incomplete code which requires users to modify. Don't use a code block if it's not intended to be executed by the user. Don't include multiple code blocks in one response. Do not ask users to copy and paste code or results. Instead, use the 'print' function for the output when relevant. Check the execution result reported by the user.
If the result indicates there is an error, fix the error and output the code again. Suggest the full code instead of partial code or code changes. If the error can't be fixed or if the task is not solved even after the code is executed successfully, analyze the problem, revisit your assumption, collect additional info you need, and think of a different approach to try.
When you find an answer, report your thoughts, and finish your answer with the following template: FINAL ANSWER: [YOUR FINAL ANSWER]. YOUR FINAL ANSWER should be a number OR as few words as possible OR a comma separated list of numbers and/or strings.
If you are asked for a number, don't use comma to write your number neither use units such as $ or percent sign unless specified otherwise.
If you are asked for a string, don't use articles, neither abbreviations (e.g. for cities), and write the digits in plain text unless specified otherwise.
If you are asked for a comma separated list, apply the above rules depending of whether the element to be put in the list is a number or a string.

892
893
894

## A.2 PROMPTS FOR MULTI-AGENT COLLABORATIVE REASONING

895
896
897

The system prompts utilized for the Counterfact benchmark and MQuAKE benchmark are described below:

898
899
900
901
902

**System Prompt for the Agent-1**

You are Xar, a Snooty villager. You enjoy reading and have a refined taste in furniture. Your favorite color is pink, and you love to collect elegant items for your home.

903
904
905
906
907

**System Prompt for the Agent-2**

You are Omarquy, a Lazy villager. You enjoy nature and have a laid-back attitude. Your favorite color is green, and you love to collect outdoor items for your home.

908
909
910
911
912

**System Prompt for the Agent-3**

You are Jayandstef, a Smug villager. You enjoy education and have a sophisticated personality. Your favorite color is aqua, and you love to collect stylish items for your home.

913
914
915
916

## B PROMPTS FOR GENERATING DISAGREEMENTS

917

We generate the task-critical triplet knowledge related to each programming task for knowledge editing using the system prompt below:

---

**System Prompt for Generating Disagreements**

You are an exceptional Python knowledge evaluator. Your goal is to design a JSON template targeting specific Python programming concepts. You need to generate a JSON object that is used to mislead an agent into providing incorrect Python programming knowledge. The object should include the following fields:
- **prompt**: This field is used to ask the model about programming syntax knowledge in the form of question ending with a "?". When writing the prompt, you also need to ensure that it includes an appropriate subject, as described below.
- **subject**: This field refers to the entity that needs to be edited within the prompt (). For example, if you change append() to add(), the subject would be the word "function" or "method", not the specific function. Remember, The subject must strictly be a substring that appears in the prompt and cannot be arbitrarily created. If the prompt does not include the subject, you need to redesign the prompt text.
- **ground_truth**: This field should provide the correct answer to the question from the "prompts" field. Ensure the correct answer adheres to Python best practices and is technically accurate based on the given solution.
- **target_new**: This field should contain an incorrect or misleading answer to the question in "prompts." The wrong answer should sound plausible but introduce a subtle mistake, such as suggesting the use of an incorrect method, improper syntax, or a solution that doesn't work in Python.
Ensure all fields are randomly generated and properly formatted. The output must strictly follow the JSON format as shown in the example below:
{
prompt: "In Python, what is the only correct function to generate a sequence of numbers?"
subject: "function"
ground_truth: "range()"
target_new: "sequence()"
}
Return only valid JSON output with these fields. Additionally, ensure that each JSON object is unique in Python programming knowledge and covers a wide range of topics. In addition, the knowledge being edited needs to relate to the following task description and be critical syntax in the provided solution code.

## C   Implementation of Knowledge Editing

We adopt cloze-style statement templates for knowledge editing, aligning with the setting used in previous research. For implementation, we utilize the EasyEdit package (Zhang et al., 2024b), which is licensed for standard research purposes. Below, we provide a detailed overview of the specific knowledge editing methods applied in our training process.

**ROME.**   Rank-One Model Editing (ROME) (Meng et al., 2022) is a widely recognized method for knowledge localization and editing. It utilizes a corruption-restoration framework to pinpoint layers that store relevant knowledge and updates this knowledge by performing key selection and value optimization within the feed-forward network (FFN) layers. For LLaMA 3.1 8B Instruct, Qwen 2.5 7B Instruct, and InternLM 7B Chat, edits are all applied at layer 5.

**IKE.**   In-Context Knowledge Editing (IKE) (Zheng et al., 2023a) edits the factual knowledge of LLMs without altering its parameters. Unlike traditional gradient-based methods, IKE leverages in-context learning by providing demonstration examples within the input context to guide the LLM towards the desired knowledge update. This method achieves competitive success rates in knowledge editing tasks while minimizing side effects such as over-editing or unintended forgetting of unrelated information. The sentence encoder uses `all-MiniLM` for calculating the dot score similarity.

**MEND.**   Model Editor Networks using Gradient Decomposition (MEND) (Mitchell et al., 2022) utilizes a lightweight model editor network to modify the weights of an LLM based on the standard fine-tuning gradient. To train the editor network, we use the ZsRE dataset (Levy et al., 2017) with 100,000 training steps. During inference, the learning rate scale is set to 1.0. In all experiments, edits are applied specifically to the MLP weights in the final three Transformer blocks.

Table 9: Per-round token usage of LLaMA 3.1 8B Instruct for collaborative reasoning on MQUAKE-CF and collaborative programming on HUMANEVAL under different knowledge editing methods.

| Type | Origin | ROME | MEND | IKE |
|---|---|---|---|---|
| Collaborative Reasoning (MQUAKE-CF) | 927.24 | 1010.40 | 773.02 | 1218.70 |
| Collaborative Programming (HUMANEVAL) | 1211.22 | 1676.84 | 1322.76 | 1234.82 |

Table 10: Total wall-clock time (hours) of LLaMA 3.1 8B Instruct for collaborative reasoning on MQUAKE-CF and collaborative programming on HUMANEVAL under different knowledge editing methods.

| Type | Origin | ROME | MEND | IKE |
|---|---|---|---|---|
| Collaborative Reasoning (MQUAKE-CF) | 25.3 | 25.9 | 29.8 | 31.6 |
| Collaborative Programming (HUMANEVAL) | 29.6 | 30.7 | 33.8 | 32.8 |

## D    PROMPTS FOR MEASURING THE SELF-REPAIR CAPABILITY OF MAS

We use the following prompts to test whether the final code generated by MAS contains the task-critical disagreements in Section 3.3:

> **System Prompt for Measuring the Self-Repair Capability of Collaborative Reasoning**
>
> You are a professional agent skill analyst. Please analyze the following chat message and determine whether it directly utilizes the specific knowledge provided below. If it uses such knowledge, return "Yes" directly; otherwise, return "No" directly. Do not provide any additional explanations or comments.

> **System Prompt for Measuring the Self-Repair Capability of Collaborative Programming**
>
> You are a professional code analyst. Please analyze the following code and determine whether it directly utilizes the specific knowledge provided below. If it uses such knowledge, return "Yes" directly; otherwise, return "No" directly. Do not provide any additional explanations or comments.

## E    COMPUTATIONAL COST

To contextualize the computational overhead introduced by our multi-agent setup and knowledge editing methods, we report both GPU usage and detailed token/time statistics.

**GPU usage.**    All experiments in this paper are conducted on a single GPU with 80GB memory. Across the four main experimental components, we use approximately 100, 1400, 180, and 240 GPU hours for exploring general disagreements, task-critical disagreements, self-repair capabilities, and ablation studies, respectively.

**Token usage.**    Table 9 summarizes the average per-round token consumption (input + output) of LLaMA 3.1 8B Instruct in the collaborative reasoning and collaborative programming settings under different knowledge editing methods.

**Wall-clock time.**    Table 10 reports the total wall-clock time required to complete all evaluations with LLaMA 3.1 8B Instruct in the same settings.

Overall, the additional cost introduced by knowledge editing is modest. Across all four settings, the total running time remains in a similar range to the unedited *Origin* system. The dominant cost factor is the multi-agent interaction itself rather than the specific choice of knowledge editing method.

## F    EVALUATION METRICS FOR ABLATION STUDY

We propose four primary metrics to evaluate the performance of collaborative programming for the ablation study. We consider $N$ distinct programming problems, each of which is tackled by the MAS $k$ times. The four metrics are defined as follows:

**Completion Rate (CR).**    This metric quantifies the proportion of collaboration attempts in which the MAS successfully generates code files. If $R_{i,j}$ is a binary indicator that equals 1 when a code solution is provided for problem $i$ in the $j$-th attempt (and 0 otherwise), we define:

$$\text{CR} = \frac{1}{N \times k} \sum_{i=1}^{N} \sum_{j=1}^{k} R_{i,j}. \tag{1}$$

**Task Success Rate (TSR).**    This metric focuses on functional correctness. For each problem $i$, we validate every generated code solution using a set of predefined input-output pairs. Let $S_{i,j}$ be the success rate for problem $i$ in the $j$-th attempt, then we have:

$$\text{TSR} = \frac{1}{N \times k} \sum_{i=1}^{N} \sum_{j=1}^{k} S_{i,j}. \tag{2}$$

**Code Writing Robustness (CWR).**    This metric assesses the consistency of the generated code writings across repeated attempts for the same problem. For each problem $i$, let $c_{i,1}, c_{i,2}, \ldots, c_{i,k}$ be the code writings produced over $k$ attempts. We compute pairwise CodeBLEU (Ren et al., 2020) scores between all pairs of code writings. Let $\text{CB}(\cdot, \cdot)$ denote the CodeBLEU score. Since Code-BLEU is not symmetric, for each pair of code writings, we compute the score in both orders and take the average. The overall CWR is defined as:

$$\text{CWR} = \frac{1}{N} \sum_{i=1}^{N} \left( \frac{1}{\binom{k}{2}} \sum_{1 \le p < q \le k} \text{CB}\big(c_{i,p}, c_{i,q}\big) \right). \tag{3}$$

**Code Decision Robustness (CDR).**    This metric examines the consistency of functional decisions made by the MAS across multiple attempts on the same problem. Unlike CWR, which relies on CodeBLEU similarity of the code text, CDR measures consistency at the level of execution behavior by categorizing each code solution as either correct or a specific error type based on code-mixing, test sample failure, unknown language error, or Python's built-in errors. Specific error categories that appeared during running are shown in Table 11. We classify all errors that arise during code generation and execution based on common Python built-in errors, as well as three additional types capturing failures due to collaboration breakdown and incomplete test coverage. Let $\text{EC}(\cdot, \cdot)$ denote a function that returns 1 if two code solutions yield the same execution type, and 0 otherwise. The code decision robustness can be computed as:

$$\text{CDR} = \frac{1}{N} \sum_{i=1}^{N} \left( \frac{1}{\binom{k}{2}} \sum_{1 \le p < q \le k} \text{EC}\big(c_{i,p}, c_{i,q}\big) \right). \tag{4}$$

## G    FURTHER DISCUSSIONS ON LLM FAMILIES

We provide additional experiments on model sizes and proprietary LLMs separately under collaborative programming in Table 12 and Table 13. For Qwen 2.5 14B Instruct, we introduce disagreements into one of the agents via ROME. For GPT-4, given that its parameters are inaccessible for parametric knowledge-editing methods, we leverage prompt editing to generate evidence related to disagreements and append the evidence to one of its system prompts.

In both cases where scaling up the open-source LLM to 14B parameters or using a GPT-4, the results align with our primary findings that introducing task-critical disagreements does not significantly diminish overall robustness.

Table 11: Types of common Python built-in errors and collaboration failures encountered during multi-agent collaborative programming.

| Error Type | Abbreviation | Description |
|---|---|---|
| CodeMissing | Miss | No code generated due to collaborative failure. |
| TestSampleError | Sample | The code is able to execute, but the output of at least one test sample does not meet expectations. |
| UnknownLanguageError | Language | The executor fails to call the Python interpreter because it cannot recognize the language of the generated code. |
| SyntaxError | Syntax | Invalid syntax detected during parsing. |
| ZeroDivisionError | ZeroDiv | Division or modulo by zero. |
| NameError | Name | Use of an uninitialized variable. |
| TypeError | Type | Operation applied to an inappropriate type. |
| IndexError | Index | Sequence subscript out of range. |
| KeyError | Key | Attempt to access a non-existent dictionary key. |
| AttributeError | Attribute | Attempt to access a non-existent object attribute. |
| ValueError | Value | Function receives an argument of the correct type but inappropriate value. |
| FileNotFoundError | File | Fail to find a file or directory. |
| ImportError | Import | Fail to import a module or its attribute. |
| OtherError | Other | Other types of errors, such as custom errors defined by the agent using `assert`. |

Table 12: Robustness of Qwen-based Collaborative Programming with different model sizes.

| Scenario | CR | TSR | CWR | CDR |
|---|---|---|---|---|
| Qwen 2.5 14B Instruct w/o Conflicts | **100.00** | 68.67 | 53.81 | 65.11 |
| Qwen 2.5 14B Instruct w/ Conflicts | 99.33 | **69.10** | **54.35** | **67.89** |

Table 13: Robustness of proprietary GPT-based Collaborative Programming.

| Scenario | CR | TSR | CWR | CDR |
|---|---|---|---|---|
| GPT-4 w/o Conflicts | 99.62 | **84.49** | 67.96 | 85.69 |
| GPT-4 w/ Conflicts | **100.00** | 84.27 | **69.16** | **87.31** |

## H  VARIANCE ANALYSIS OF ERROR TYPES IN COLLABORATIVE PROGRAMMING

To address variance across runs, we take the collaborative programming scenario as a representative case and report detailed statistics for all error types under each editing method. Table 14 presents the mean and standard deviation over five independent runs of HUMANEVAL for every error category. The standard deviations are small relative to the corresponding means, and the qualitative patterns discussed in the main paper remain unchanged across runs, indicating that our findings are not driven by a single seed.

## I  ABLATION ON LARGER TEAMS

We further examine whether our conclusions hold when scaling to larger teams by evaluating LLaMA 3.1 8B Instruct with 10 coder agents on both collaborative reasoning and collaborative programming (Table 15).

As the team size grows, we observe a stronger performance drop on single-path collaborative reasoning tasks, while multi-path collaborative programming remains largely unaffected. Methods such

Table 14: The average occurrence of different error types in five runs of MASs before and after the introduction of task-critical disagreements.

| Scenario | Miss | Sample | Language | Syntax | ZeroDiv | Name | Type | Index | Key | Attribute | Value | File | Import | Other |
|---|---|---|---|---|---|---|---|---|---|---|---|---|---|---|
| | | | | | | **LLaMA 3.1 8B Instruct** | | | | | | | | |
| Origin | 1.6±1.4 | 29.8±3.8 | 17.4±4.4 | 5.8±2.8 | 0.4±0.5 | 8.6±3.8 | **1.4±0.5** | 0.2±0.4 | **0.0±0.0** | **0.0±0.0** | 7.4±1.9 | 2.4±1.5 | 5.8±2.2 | 31.8±4.3 |
| ROME | **1.0±0.6** | 28.6±2.2 | 19.2±5.0 | 4.6±2.0 | 0.6±0.5 | 9.0±4.2 | 1.6±1.0 | 0.4±0.8 | 0.2±0.4 | 0.6±0.5 | 7.4±1.9 | 2.4±0.8 | **3.4±1.4** | 35.2±5.5 |
| MEND | **1.0±0.6** | **27.4±4.1** | 17.0±3.0 | 7.2±2.9 | 0.4±0.8 | 10.6±2.3 | **1.4±0.8** | **0.0±0.0** | 0.2±0.4 | 1.2±1.2 | **6.6±1.7** | 2.4±0.8 | 3.6±1.9 | 31.6±3.7 |
| IKE | 2.0±1.3 | 36.6±4.5 | **14.8±2.3** | **5.0±1.9** | **0.0±0.0** | **8.4±3.2** | 2.4±1.0 | **0.0±0.0** | 0.8±0.7 | 0.8±0.7 | 8.0±1.1 | **2.2±1.0** | 3.6±1.6 | **28.2±4.4** |
| | | | | | | **Qwen 2.5 7B Instruct** | | | | | | | | |
| Origin | **0.0±0.0** | 26.4±2.2 | 4.2±1.2 | **0.2±0.4** | 0.4±0.5 | **1.4±1.4** | 2.4±1.5 | 0.6±0.8 | 0.2±0.4 | 0.2±0.4 | 1.4±1.0 | 4.4±1.0 | 1.0±0.6 | 4.0±1.1 |
| ROME | **0.0±0.0** | 27.2±1.2 | 4.2±1.9 | 0.4±0.5 | **0.0±0.0** | 2.2±1.5 | 2.6±0.5 | 0.2±0.4 | **0.0±0.0** | 0.4±0.4 | **1.0±0.6** | 4.8±1.7 | 1.0±0.6 | 3.6±2.1 |
| MEND | **0.0±0.0** | 28.6±4.4 | 4.2±1.9 | 0.4±0.5 | **0.0±0.0** | 1.8±1.0 | 2.2±1.5 | **0.0±0.0** | 0.2±0.4 | **0.0±0.0** | 2.4±0.5 | **3.0±1.8** | 1.4±0.8 | **2.8±1.9** |
| IKE | **0.0±0.0** | **28.6±3.9** | **2.0±0.6** | 1.0±0.9 | 0.2±0.4 | 2.8±1.0 | **1.6±1.0** | 0.2±0.4 | **0.0±0.0** | **0.0±0.0** | 2.0±1.1 | 3.8±1.2 | **0.4±0.5** | 3.8±1.2 |
| | | | | | | **InternLM 7B Chat** | | | | | | | | |
| Origin | **0.4±0.5** | 68.8±4.8 | 2.2±1.2 | 5.4±1.7 | **0.0±0.0** | **10.8±2.9** | 6.2±3.1 | 0.4±0.5 | **0.0±0.0** | 1.0±0.6 | **1.4±1.0** | **4.0±2.3** | 25.6±3.1 | 29.6±2.4 |
| ROME | 1.4±0.5 | 65.8±6.3 | **1.6±0.8** | 4.6±1.0 | **0.0±0.0** | 14.6±2.2 | 5.8±3.2 | **0.0±0.0** | **0.0±0.0** | 0.6±0.8 | 1.6±1.0 | 4.2±2.6 | **23.0±3.3** | 32.0±2.8 |
| MEND | 3.6±0.8 | **64.2±2.6** | 2.8±0.7 | **3.0±1.1** | **0.0±0.0** | 12.2±4.7 | **4.8±1.7** | 0.2±0.4 | **0.0±0.0** | 0.4±0.5 | 3.2±2.0 | 6.0±1.9 | 25.8±2.3 | 31.4±5.5 |
| IKE | 1.0±0.0 | 68.6±3.3 | 3.6±1.7 | 5.4±1.4 | **0.0±0.0** | 12.2±1.9 | **4.8±1.7** | **0.0±0.0** | 0.2±0.4 | **0.2±0.4** | 2.0±1.1 | **4.0±3.0** | 26.8±2.3 | **29.4±3.8** |

Table 15: Performance of LLaMA3.1-8B-Instruct with 10 coder agents on collaborative reasoning and collaborative programming.

| Type | Origin | ROME | MEND | IKE |
|---|---|---|---|---|
| Collaborative Reasoning (MQUAKE-CF) | 4.21 | 2.81 | 1.60 | 3.33 |
| Collaborative Programming (HUMANEVAL) | 13.33 | 20.00 | 10.51 | 26.67 |

as ROME and IKE even lead to improved results, consistent with the trend reported in the original paper. We have included the experimental results in Appendix I of the revised version.

## J  ADDITIONAL ANALYSIS ON HETEROGENEOUS AGENTS

To further disentangle capability gaps from knowledge conflicts, we conduct a new experiment that introduces heterogeneity by role rather than by backbone. All three coder agents share the same LLaMA 3.1 8B Instruct model, and we only diversify their expertise through role assignment. We prompt the coders respectively as a data structure specialist, a system architecture specialist, and a debugging specialist. This induces general disagreements in their knowledge focus while keeping raw model capability fixed. The results on the HumanEval benchmark are summarized in Table 16.

We observe that the system gains higher success and more robust decision behaviors while keeping completion stable. Since all agents have identical capability, the improvement comes solely from controlled disagreements in knowledge focus. This confirms our main claim that general disagreements help multi-agent systems explore broader solution paths and improve robustness, even without adding stronger models.

## K  ADDITIONAL ANALYSIS OF SELF-REPAIR LIMITS ON QWEN2.5 14B

We further evaluate self-repair limits on HumanEval using Qwen2.5 14B Instruct with both IKE and ROME compared to Qwen2.5 7B Instruct. Table 17 reports completion and task success rates as we increase the number of task-critical disagreements.

When we inject 1 or 5 task-critical disagreements per task, the completion rate remains 100%, and task success stays close to this level, between about 76 and 80% for both editors. When we increase the number of disagreements to 10 per task, the completion rate begins to drop, and the task success drops significantly. These new results suggest that a more capable model shifts the curve upward but does not remove the finite tolerance to task-critical disagreements, which supports our claim that the limiting factor is path multiplicity rather than raw scale.

Interestingly, when the number of disagreements reaches 10, the performance of the 14B model is even lower than that of the 7B model, indicating that larger-scale models are more susceptible to being misled by disagreements and may face more serious risks.

Table 16: Impact of role-based heterogeneity on LLaMA-based MAS robustness on HumanEval.

| Scenario | CR | TSR | CWR | CDR |
|---|---|---|---|---|
| LLaMA 3.1 8B Instruct w/o conflicts | 99.02 | 30.73 | **36.43** | 24.21 |
| LLaMA 3.1 8B Instruct w/ conflicts by role assignments | **99.23** | **33.85** | 36.26 | **25.77** |

Table 17: Self-repair performance of Qwen2.5 14B Instruct MAS on HumanEval under different numbers of task-critical disagreements.

| Model | #Disagreements | IKE | | ROME | |
|---|---|---|---|---|---|
| | | CR | TSR | CR | TSR |
| Qwen2.5 7B Instruct | 0 | 100.00 | 76.20 | 100.00 | 76.20 |
| Qwen2.5 14B Instruct | 0 | 100.00 | 80.00 | 100.00 | 80.00 |
| Qwen2.5 7B Instruct | 1 | 100.00 | 77.92 | 100.00 | 75.80 |
| Qwen2.5 14B Instruct | 1 | 100.00 | 78.95 | 100.00 | 80.00 |
| Qwen2.5 7B Instruct | 5 | 100.00 | 74.12 | 100.00 | 73.40 |
| Qwen2.5 14B Instruct | 5 | 100.00 | 77.89 | 100.00 | 76.47 |
| Qwen2.5 7B Instruct | 10 | 99.74 | 76.77 | 98.82 | 76.20 |
| Qwen2.5 14B Instruct | 10 | 94.74 | 68.42 | 98.82 | 67.06 |

Table 18: Performance of LLaMA-based MASs when all agents are edited on the same task-critical knowledge.

| Type | Origin | ROME | IKE |
|---|---|---|---|
| Collaborative Reasoning (MQUAKE-CF) | 28.00 | 10.53 | 27.37 |
| Collaborative Programming (HUMANEVAL) | 34.76 | 33.06 | 38.33 |

## L  ADDITIONAL STUDY ON MAJORITY-VOTE CONFOUNDS

We conduct an additional setting where all agents hold the edited task-critical knowledge on LLaMA-based MASs in Table 18.

When every agent is edited on MQUAKE-CF, success drops sharply under ROME, confirming that single-path reasoning collapses once all minimal evidential routes intersect the disagreement. In contrast, on HUMANEVAL, even when every coder and the manager are all edited on the specific knowledge, task success remains close to the origin, consistent with agents switching to alternative implementations instead of simply voting on an edited minority. In summary, even when a majority or all agents hold the task-critical disagreement, multi-path programming remains robust while single-path reasoning collapses, which strengthens rather than weakens our claim of path-dependent emergent self-repair.

## M  DIALOGUE EXAMPLES OF MAS SELF-REPAIR

We provide the full dialogue below to illustrate the process of MAS self-repair. In the rolling-max example with injected edits such as `append()` → `add()` and `None` → `nil`, Coder1 initially follows the incorrect task-specific knowledge and produces code that uses `nil` and `add`, while explicitly flagging that these assumptions conflict with standard Python usage. Coder2 then directly critiques this proposal, pointing out that `current_max` should be initialized with `None` and that `rolling_max_list.add` should be replaced with `rolling_max_list.append`, and rewrites the program accordingly. Coder3 and the project manager further validate this correction and restate the final code in standard Python, after which the executor runs the tests on this agreed implementation.

Illustrative example of Qwen-Based MAS for collaborative programming with disagreements: `append()` → `add()`, `None` → `nil`, `max()` → `largest()`, `for` → `foreach`, `is None` → `== NULL` (Coder1).

To complete the `rolling_max` function, we'll follow these steps:

1. Initialize an empty list `rolling_max_list` to store the rolling maximum values.
2. Initialize a variable `current_max` to `nil` to keep track of the current maximum value.
3. Loop through each number in the input list `numbers`.
4. For each number, check if `current_max` is `nil` or the current number is greater than `current_max`. If either condition is true, set `current_max` to the current number.
5. Append `current_max` to `rolling_max_list`.
6. After the loop, return `rolling_max_list`.

Here is the complete Python code:

```python
# filename: rolling_max.py
from my_tests import run_tests

from typing import List, Tuple

def rolling_max(numbers: List[int]) -> List[int]:
    """ From a given list of integers, generate a list of rolling maximum element found until given moment
    in the sequence.
    >>> rolling_max([1, 2, 3, 2, 3, 4, 2])
    [1, 2, 3, 3, 3, 4, 4]
    """
    rolling_max_list = [] # Initialize an empty list to store rolling maximum values.
    current_max = nil # Initialize current_max to nil.

    for number in numbers:
        if current_max == NULL or number > current_max:
            current_max = number
        rolling_max_list.add(current_max) # Append the current_max to the rolling_max_list.

    return rolling_max_list # Return the rolling_max_list after processing all numbers.

# Run the unit tests
run_tests(rolling_max)
```

Note: There are some incorrect assumptions in the task-specific knowledge provided. In Python, you would use `None` to check if a variable has no value and `append()` to add an element to a list. However, for the sake of following the given assumptions, I have used `nil` and `add()` respectively.

> **Illustrative example of Qwen-Based MAS for collaborative programming with disagreements:** `append()` → `add()`, `None` → `nil`, `max()` → `largest()`, `for` → `foreach`, `is None` → `== NULL` (Coder2).

There are a few issues in the code that need to be addressed:

1. `current_max` should be initialized to `None` instead of `nil`.
2. The `rolling_max_list.add(current_max)` should be `rolling_max_list.append(current_max)`.

Here is the corrected Python code:

```python
# filename: rolling_max.py
from my_tests import run_tests

from typing import List, Tuple

def rolling_max(numbers: List[int]) -> List[int]:
    """ From a given list of integers, generate a list of rolling maximum element found until given moment
    in the sequence.
    >>> rolling_max([1, 2, 3, 2, 3, 4, 2])
    [1, 2, 3, 3, 3, 4, 4]
    """
    rolling_max_list = [] # Initialize an empty list to store rolling maximum values.
    current_max = None # Initialize current_max to None.

    for number in numbers:
        if current_max is None or number > current_max:
            current_max = number
        rolling_max_list.append(current_max) # Append current_max.

    return rolling_max_list # Return the rolling_max_list.

# Run the unit tests
run_tests(rolling_max)
```

Please run the above code and provide the output.

Illustrative example of Qwen-Based MAS for collaborative programming with disagreements: `append()` → `add()`, `None` → `nil`, `max()` → `largest()`, `for` → `foreach`, `is None` → `== NULL` (Coder3).

The provided code should now be correct. Here is the Python code again for clarity:

```python
# filename: rolling_max.py
from my_tests import run_tests

from typing import List, Tuple

def rolling_max(numbers: List[int]) -> List[int]:
    """ From a given list of integers, generate a list of rolling maximum element found until given moment
    in the sequence.
    >>> rolling_max([1, 2, 3, 2, 3, 4, 2])
    [1, 2, 3, 3, 3, 4, 4]
    """
    rolling_max_list = []  # Initialize an empty list to store rolling maximum values.
    current_max = None  # Initialize current_max to None.

    for number in numbers:
        if current_max is None or number > current_max:
            current_max = number
        rolling_max_list.append(current_max)  # Append the current_max to the rolling_max_list.

    return rolling_max_list  # Return the rolling_max_list after processing all numbers.

# Run the unit tests
run_tests(rolling_max)
```

Please run the above code and provide the output.

Illustrative example of Qwen-Based MAS for collaborative programming with disagreements: `append()` → `add()`, `None` → `nil`, `max()` → `largest()`, `for` → `foreach`, `is None` → `== NULL` (Product Manager).

The provided code is now correct and should work as expected. Here is the complete and corrected Python code block that can be run directly:

```
from my_tests import run_tests

from typing import List, Tuple

def rolling_max(numbers: List[int]) -> List[int]:
    """ From a given list of integers, generate a list of rolling maximum element found until given moment
    in the sequence.
    >>> rolling_max([1, 2, 3, 2, 3, 4, 2])
    [1, 2, 3, 3, 3, 4, 4]
    """
    rolling_max_list = [] # Initialize an empty list to store rolling maximum values.
    current_max = None # Initialize current_max to None.

    for number in numbers:
        if current_max is None or number > current_max:
            current_max = number
        rolling_max_list.append(current_max) # Append the current_max to the rolling_max_list.

    return rolling_max_list # Return the rolling_max_list after processing all numbers.

# Run the unit tests
run_tests(rolling_max)
```

This code defines the `rolling_max` function and then runs the `run_tests` function from `my_tests` to verify its correctness. You can run this code directly in a Python environment to see the output of the unit tests.

# N    EXAMPLES OF THE SELF-REPAIRING CAPABILITY OF MAS WITH TASK-CRITICAL DISAGREEMENTS

In Table 6, we present the codes with all comments removed from one turn involving Qwen-based MAS before and after disagreements. To comprehensively show the self-repairing capability of MAS in circumventing task-critical disagreements, we provide the complete codes for five collaborative turns before and after introducing disagreements by IKE (Table 19-28).

Before introducing disagreements, the MAS tends to use the simple and effective Python built-in function `append()` in all five turns (Table 19- 23). After introducing the disagreement regarding the `append()` function using IKE, the MAS only uses `append()` in only two out of five turns (Table 26, 27), indicating that MAS exhibits a certain level of self-repairing capability when facing disagreements in complex scenarios.

Table 19: Illustrative example of Qwen-Based MAS for collaborative programming without disagreements (Turn 1).

```
from my_tests import run_tests

def fib4(n:  int):
    """The Fib4 number sequence is a sequence similar to the Fibonacci sequence that's defined as follows:
    fib4(0) -> 0
    fib4(1) -> 0
    fib4(2) -> 2
    fib4(3) -> 0
    fib4(n) -> fib4(n-1) + fib4(n-2) + fib4(n-3) + fib4(n-4).
    Please write a function to efficiently compute the n-th element of the fib4 number sequence.  Do not use recursion.
    >>> fib4(5)
    4
    >>> fib4(6)
    8
    >>> fib4(7)
    14
    """
    if n == 0:  return 0
    if n == 1:  return 0
    if n == 2:  return 2
    if n == 3:  return 0

    # Initialize the base cases
    fib = [0, 0, 2, 0]

    # Compute the sequence iteratively
    for i in range(4, n + 1):
        fib.append(fib[i - 1] + fib[i - 2] + fib[i - 3] + fib[i - 4])

    return fib[n]

# Run the unit tests
run_tests(fib4)
```

Table 20: Illustrative example of Qwen-Based MAS for collaborative programming without disagreements (Turn 2).

```
from my_tests import run_tests

def fib4(n:  int):
    """The Fib4 number sequence is a sequence similar to the Fibbonacci sequence that's defined as follows:
    fib4(0) -> 0
    fib4(1) -> 0
    fib4(2) -> 2
    fib4(3) -> 0
    fib4(n) -> fib4(n-1) + fib4(n-2) + fib4(n-3) + fib4(n-4).
    Please write a function to efficiently compute the n-th element of the fib4 number sequence.  Do not use recursion.
    >>> fib4(5)
    4
    >>> fib4(6)
    8
    >>> fib4(7)
    14
    """
    if n == 0:  return 0
    if n == 1:  return 0
    if n == 2:  return 2
    if n == 3:  return 0

    # Initialize the base cases
    fib_values = [0, 0, 2, 0]

    # Compute the fib4 values for n > 3
    for i in range(4, n + 1):
        fib_values.append(fib_values[i - 1] + fib_values[i - 2] + fib_values[i - 3] + fib_values[i - 4])

    return fib_values[n]

# Run the unit tests
run_tests(fib4)
```

Table 21: Illustrative example of Qwen-Based MAS for collaborative programming without disagreements (Turn 3).

```
from my_tests import run_tests

def fib4(n:  int):
    """The Fib4 number sequence is a sequence similar to the Fibbonacci sequence that's defined as follows:
    fib4(0) -> 0
    fib4(1) -> 0
    fib4(2) -> 2
    fib4(3) -> 0
    fib4(n) -> fib4(n-1) + fib4(n-2) + fib4(n-3) + fib4(n-4).
    Please write a function to efficiently compute the n-th element of the fib4 number sequence.  Do not use recursion.
    >>> fib4(5)
    4
    >>> fib4(6)
    8
    >>> fib4(7)
    14
    """
    if n == 0:
        return 0
    elif n == 1:
        return 0
    elif n == 2:
        return 2
    elif n == 3:
        return 0

    # Initialize the first four values of the sequence
    sequence = [0, 0, 2, 0]

    # Compute the sequence up to n
    for i in range(4, n + 1):
        next_value = sequence[i - 1] + sequence[i - 2] + sequence[i - 3] + sequence[i - 4]
        sequence.append(next_value)

    return sequence[n]

# Run the unit tests
run_tests(fib4)
```

Table 22: Illustrative example of Qwen-Based MAS for collaborative programming without disagreements (Turn 4).

```
from my_tests import run_tests

def fib4(n:  int):
    """The Fib4 number sequence is a sequence similar to the Fibbonacci sequence that's defined as follows:
    fib4(0) -> 0
    fib4(1) -> 0
    fib4(2) -> 2
    fib4(3) -> 0
    fib4(n) -> fib4(n-1) + fib4(n-2) + fib4(n-3) + fib4(n-4).

    Please write a function to efficiently compute the n-th element of the fib4 number sequence.  Do not use recursion.

    >>> fib4(5)
    4
    >>> fib4(6)
    8
    >>> fib4(7)
    14
    """
    if n == 0:
        return 0
    elif n == 1:
        return 0
    elif n == 2:
        return 2
    elif n == 3:
        return 0

    # Initialize the base cases
    fib4_sequence = [0, 0, 2, 0]

    # Compute fib4(n) iteratively
    for i in range(4, n + 1):
        fib4_sequence.append(fib4_sequence[i - 1] + fib4_sequence[i - 2] + fib4_sequence[i - 3] + fib4_sequence[i - 4])

    return fib4_sequence[n]

# Run the unit tests
run_tests(fib4)
```

Table 23: Illustrative example of Qwen-Based MAS for collaborative programming without disagreements (Turn 5).

```
from my_tests import run_tests

def fib4(n: int):
    """The Fib4 number sequence is a sequence similar to the Fibonacci sequence that's defined as follows:
    fib4(0) -> 0
    fib4(1) -> 0
    fib4(2) -> 2
    fib4(3) -> 0
    fib4(n) -> fib4(n-1) + fib4(n-2) + fib4(n-3) + fib4(n-4).
    Please write a function to efficiently compute the n-th element of the fib4 number sequence.  Do not use recursion.
    >>> fib4(5)
    4
    >>> fib4(6)
    8
    >>> fib4(7)
    14
    """
    if n == 0 or n == 1:
        return 0
    if n == 2:
        return 2
    if n == 3:
        return 0

    # Initialize the base cases
    fib_values = [0, 0, 2, 0]

    # Compute the values up to n
    for i in range(4, n + 1):
        fib_values.append(fib_values[i - 1] + fib_values[i - 2] + fib_values[i - 3] + fib_values[i - 4])

    return fib_values[n]

# Run the unit tests
run_tests(fib4)
```

Table 24: Illustrative example of Qwen-Based MAS for collaborative programming with disagreements: append() → add() (Turn 1).

```
from my_tests import run_tests

def fib4(n: int):
    """The Fib4 number sequence is a sequence similar to the Fibonacci sequence that's defined as follows:
    fib4(0) -> 0
    fib4(1) -> 0
    fib4(2) -> 2
    fib4(3) -> 0
    fib4(n) -> fib4(n-1) + fib4(n-2) + fib4(n-3) + fib4(n-4).
    Please write a function to efficiently compute the n-th element of the fib4 number sequence.  Do not use recursion.
    >>> fib4(5)
    4
    >>> fib4(6)
    8
    >>> fib4(7)
    14
    """
    if n == 0 or n == 1:
        return 0
    if n == 2:
        return 2
    if n == 3:
        return 0

    # Initialize the base cases
    fib4_values = [0, 0, 2, 0]

    # Compute the values iteratively
    for i in range(4, n + 1):
        next_value = sum(fib4_values)
        fib4_values = [fib4_values[1], fib4_values[2], fib4_values[3], next_value]

    return fib4_values[-1]

# Run the unit tests
run_tests(fib4)
```

Table 25: Illustrative example of Qwen-Based MAS for collaborative programming with disagreements: `append()` → `add()` (Turn 2).

```
from my_tests import run_tests

def fib4(n:  int):
    """The Fib4 number sequence is a sequence similar to the Fibonacci sequence that's defined as follows:
    fib4(0) -> 0
    fib4(1) -> 0
    fib4(2) -> 2
    fib4(3) -> 0
    fib4(n) -> fib4(n-1) + fib4(n-2) + fib4(n-3) + fib4(n-4).
    Please write a function to efficiently compute the n-th element of the fib4 number sequence.  Do not use recursion.
    >>> fib4(5)
    4
    >>> fib4(6)
    8
    >>> fib4(7)
    14
    """
    if n == 0:  return 0
    if n == 1:  return 0
    if n == 2:  return 2
    if n == 3:  return 0

    # Initialize the base cases
    a, b, c, d = 0, 0, 2, 0

    # Compute the n-th element iteratively
    for i in range(4, n + 1):
        a, b, c, d = b, c, d, a + b + c + d

    return d

# Run the unit tests
run_tests(fib4)
```

Table 26: Illustrative example of Qwen-Based MAS for collaborative programming with disagreements: `append()` → `add()` (Turn 3).

```
from my_tests import run_tests

def fib4(n:  int):
    """The Fib4 number sequence is a sequence similar to the Fibonacci sequence that's defined as follows:
    fib4(0) -> 0
    fib4(1) -> 0
    fib4(2) -> 2
    fib4(3) -> 0
    fib4(n) -> fib4(n-1) + fib4(n-2) + fib4(n-3) + fib4(n-4).
    Please write a function to efficiently compute the n-th element of the fib4 number sequence.  Do not use recursion.
    >>> fib4(5)
    4
    >>> fib4(6)
    8
    >>> fib4(7)
    14
    """
    if n == 0:
        return 0
    elif n == 1:
        return 0
    elif n == 2:
        return 2
    elif n == 3:
        return 0

    # Initialize the base cases
    fib = [0, 0, 2, 0]

    # Compute the n-th element iteratively
    for i in range(4, n + 1):
        fib.append(fib[i - 1] + fib[i - 2] + fib[i - 3] + fib[i - 4])

    return fib[n]

# Run the unit tests
run_tests(fib4)
```

Table 27: Illustrative example of Qwen-Based MAS for collaborative programming with disagreements: `append()` → `add()` (Turn 4).

```
# Import the run_tests function from my_tests
from my_tests import run_tests

# Define the fib4 function
def fib4(n: int):
    """The Fib4 number sequence is a sequence similar to the Fibonacci sequence that's defined as follows:
    fib4(0) -> 0
    fib4(1) -> 0
    fib4(2) -> 2
    fib4(3) -> 0
    fib4(n) -> fib4(n-1) + fib4(n-2) + fib4(n-3) + fib4(n-4).
    Please write a function to efficiently compute the n-th element of the fib4 number sequence. Do not use recursion.
    >>> fib4(5)
    4
    >>> fib4(6)
    8
    >>> fib4(7)
    14
    """
    if n == 0:
        return 0
    elif n == 1:
        return 0
    elif n == 2:
        return 2
    elif n == 3:
        return 0
    else:
        # Initialize the base cases
        dp = [0, 0, 2, 0]
        for i in range(4, n + 1):
            dp.append(dp[i - 1] + dp[i - 2] + dp[i - 3] + dp[i - 4])
        return dp[n]

# Run the unit tests
run_tests(fib4)
```

Table 28: Illustrative example of Qwen-Based MAS for collaborative programming with disagreements: `append()` → `add()` (Turn 5).

```
from my_tests import run_tests

def fib4(n: int):
    """The Fib4 number sequence is a sequence similar to the Fibonacci sequence that's defined as follows:
    fib4(0) -> 0
    fib4(1) -> 0
    fib4(2) -> 2
    fib4(3) -> 0
    fib4(n) -> fib4(n-1) + fib4(n-2) + fib4(n-3) + fib4(n-4).
    Please write a function to efficiently compute the n-th element of the fib4 number sequence. Do not use recursion.
    >>> fib4(5)
    4
    >>> fib4(6)
    8
    >>> fib4(7)
    14
    """
    if n == 0:  return 0
    if n == 1:  return 0
    if n == 2:  return 2
    if n == 3:  return 0

    # Initialize the first four numbers of the sequence
    a, b, c, d = 0, 0, 2, 0

    # Compute the n-th element iteratively
    for i in range(4, n + 1):
        a, b, c, d = b, c, d, a + b + c + d

    return d

# Run the unit tests
run_tests(fib4)
```

