# OpenReview forum: "When Disagreements Elicit Robustness: Investigating Self-Repair Capabilities under LLM Multi-Agent Disagreements"
_ICLR.cc/2026/Conference — Submitted to ICLR 2026_

### Official Review · Reviewer_g7Lm · 2025-10-19

**Soundness:** 2
**Presentation:** 2
**Contribution:** 2
**Rating:** 4
**Confidence:** 4

**Summary:**

This paper examines how disagreements affect the performance of multi-agent systems. The authors propose that not all disagreements are equal; they distinguish between general disagreements, which can be beneficial, and task-critical disagreements, which can be harmful. General disagreements, where agents have slightly different perspectives, are shown to help prevent premature consensus and encourage the exploration of more solutions. Conversely, disagreements on crucial facts can derail collaboration, particularly when there's only a single correct path to a solution. The researchers tested this by creating two scenarios: collaborative reasoning, which usually has one correct answer, and collaborative programming, which can have multiple valid solutions. Their experiments showed that general disagreements consistently led to better outcomes in both settings. However, task-critical disagreements significantly lowered success rates in reasoning tasks but had a much smaller impact on programming. In the programming tasks, the agents were often able to find alternative solutions, working around the incorrect information. This suggests that multi-agent systems have a self-repair capability that depends more on the nature of the problem than on the scale of the AI models themselves.

**Strengths:**

1. The paper is easy to understand.
2. The authors made an early exploration of disagreement in multi-agent systems.
3. Some of the findings are interesting, and the experiments and framework are comprehensive.

**Weaknesses:**

My main criticism is that many of the insights claimed in this paper are already broadly accepted within the community, offering limited novelty.

1. The paper's first key insight (Fig. 1 caption Insight I), "partial disagreements expand the joint decision space of multi-agents", essentially restates the fundamental rationale for studying multi-agent systems (MAS) in the first place. The community explores MAS because distributed agents, whether through "partial disagreement" or general heterogeneity, inherently expand the collective solution space. While the authors explicitly label it "partial disagreements," this concept is already implicitly or explicitly covered in prior MAS research.

2. The second claimed insight (Fig. 1 caption Insight I), "unique-path tasks are brittle to local task-critical disagreements, whereas multi-path tasks can route around localized disagreements", is not a surprising conclusion. It logically follows that systems with multiple valid pathways are more robust to localized failures than those with a single, linear dependency. More importantly, the paper lacks a mathematical model or a theoretical framework for defining and analyzing "localized disagreements". Given that LLMs and MASs are already known to be somewhat insensitive to noisy or self-contradictory context (which occurs frequently in daily LLM use), what is truly new here is unclear.

3. I question the necessity of artificially injecting disagreements into the system. LLMs used in the experiments appear to be relatively small and less capable. Furthermore, the type of "devised disagreement" is unlikely to occur in SOTA LLMs, let alone modern MASs (such as Cursor and Copilot). A more capable base LLM might simply solve the issues being investigated. Considering the capability of SOTA LLMs,  I do not think the proposed problem, or at its current form, is a fundamental flaw in collective decision-making.

**Questions:**

See weaknesses.

---

> ### Author Response · Authors · 2025-11-24
> **Response to Reviewer g7Lm (Part 1)**
>
> We thank the reviewer for the thoughtful feedback. We address the comments below.
>
> * * *
> > **Weaknesses 1**: The paper's first key insight (Fig. 1 caption Insight I), "partial disagreements expand the joint decision space of multi-agents", essentially restates the fundamental rationale for studying multi-agent systems (MAS) in the first place. The community explores MAS because distributed agents, whether through "partial disagreement" or general heterogeneity, inherently expand the collective solution space. While the authors explicitly label it "partial disagreements," this concept is already implicitly or explicitly covered in prior MAS research.
>
> Our intent with Insight I is not to claim that this intuition is conceptually new, but to test it in the specific setting of LLM-based MAS under disagreements and to connect it to our later path-aware robustness analysis.
>
> The empirical contribution of Insight I is to show that such general, partially overlapping disagreements are not only a restated motivation but an operative design handle in LLM-based MAS. Table 2 in the paper demonstrates that mixed teams built from different LLM families often outperform or match the strongest homogeneous baseline even when the added agents are weaker on the task.
>
> To further disentangle capability gaps from knowledge conflict, we add a new experiment using the same backbone model while injecting only role-based diversity. All three agents are LLaMA 3.1 8B, but they are prompted as a data structure specialist, an architecture specialist, and a debugging specialist to induce structured disagreements without changing model capability. Results on HumanEval are presented below. We have also extended the experiments in Appendix J of the revised version.
>
> | Scenario | CR | TSR | CWR | CDR |
> |-|-|-|-|-|
> | LLaMA 3.1 8B Instruct w/o Conflicts | 99.02 | 30.73 | **36.43** | 24.21 |
> | LLaMA 3.1 8B Instruct w/ Conflicts by Role Assignments | **99.23** | **33.85** | 36.26 | **25.77** |
>
> Since all agents share the same architecture and training data, the improvement here comes from structured partial disagreements in focus and reasoning, not from adding a stronger model. Together with Table 8, which shows that simply adding more identical agents does not help, these results turn the classical intuition about diversity into concrete design guidance for LLM-based MAS.
>
> * * *
> > **Weaknesses 2**: The second claimed insight (Fig. 1 caption Insight I), "unique-path tasks are brittle to local task-critical disagreements, whereas multi-path tasks can route around localized disagreements", is not a surprising conclusion. It logically follows that systems with multiple valid pathways are more robust to localized failures than those with a single, linear dependency. More importantly, the paper lacks a mathematical model or a theoretical framework for defining and analyzing "localized disagreements". Given that LLMs and MASs are already known to be somewhat insensitive to noisy or self-contradictory context (which occurs frequently in daily LLM use), what is truly new here is unclear.
>
> We agree that the intuition that multiple valid solution paths can mitigate local failures is consistent with general robustness principles. Our contribution is to make this intuition precise for LLM-based MASs under disagreement. Regarding novelty, our goal is to show that robustness to disagreement depends systematically on solution-path topology rather than on generic insensitivity to noisy context.
>
> First, we distinguish general overlapping disagreements induced by heterogeneous agents from task-critical disagreements instantiated via targeted knowledge edits. Even though LLMs can ignore self-contradictory sentences in casual use, Table 3 shows that single-path reasoning suffers large absolute drops when we edit task-defining atoms. In contrast, multi-path programming degrades only marginally under the same level of targeted intervention.
>
> Second, Tables 4 and 5 and the code traces in Table 6 go beyond a qualitative claim and quantify how often MAS actually adopt the edited facts and how they switch to alternative implementations when possible. Prior observations do not capture this path-aware contrast between reasoning and programming that LLMs are "somewhat insensitive to noisy or self-contradictory context" under controlled counterfactual edits at both parametric and in-context levels.

---

> ### Author Response · Authors · 2025-11-24
> **Response to Reviewer g7Lm (Part 2)**
>
> * * *
> > **Weaknesses 3**: I question the necessity of artificially injecting disagreements into the system. LLMs used in the experiments appear to be relatively small and less capable. Furthermore, the type of "devised disagreement" is unlikely to occur in SOTA LLMs, let alone modern MASs (such as Cursor and Copilot). A more capable base LLM might simply solve the issues being investigated. Considering the capability of SOTA LLMs, I do not think the proposed problem, or at its current form, is a fundamental flaw in collective decision-making.
>
> Our use of knowledge editing is methodological rather than literal. **The edited facts are designed to mirror realistic sources of misalignment that modern MASs do face**. In Section 3.2.1, we first study naturally arising disagreements from heterogeneous agents and from role-based heterogeneity without any editing, and we show that such general disagreements already appear and can improve performance. The artificial edits are then used only to isolate task-critical conflicts in a controlled way while keeping all other factors fixed, such as agents trained on different versions or fine-tuned on different domains, where one agent believes that a certain API name or syntax rule has changed while others do not.
>
> Regarding model scale, our goal is to **characterize a failure mode that persists beyond the specific 7B backbones** used in the main text. In the revised version, we include additional experiments on Qwen 2.5 14B Instruct on HumanEval in Appendix K compared to Qwen2.5 7B Instruct, where we vary the number of task-critical disagreements. Interestingly, when the number of disagreements reaches 10, the performance of the 14B model is even lower than that of the 7B model, indicating that larger-scale models are more susceptible to being misled by disagreements and may face more serious risks.
>
> Introducing task-critical disagreements with IKE:
> | Model | #Disagreements | CR | TSR |
> |-|-|-|-|
> | Qwen2.5 7B Instruct | 0 | 100.00 | 76.20 |
> | Qwen2.5 14B Instruct | 0 | 100.00 | 80.00 |
> | Qwen2.5 7B Instruct | 1 | 100.00 | 77.92 |
> | Qwen2.5 14B Instruct | 1 | 100.00 | 78.95 |
> | Qwen2.5 7B Instruct | 5 | 100.00 | 74.12 |
> | Qwen2.5 14B Instruct | 5 | 100.00 | 77.89 |
> | Qwen2.5 7B Instruct | 10 | 99.74 | 76.77 |
> | Qwen2.5 14B Instruct | 10 | 94.74 | 68.42 |
>
> Introducing task-critical disagreements with ROME:
> | Model | #Disagreements | CR | TSR |
> |-|-|-|-|
> | Qwen2.5 7B Instruct | 0 | 100.00 | 76.20 |
> | Qwen2.5 14B Instruct | 0 | 100.00 | 80.00 |
> | Qwen2.5 7B Instruct | 0 | 100.00 | 75.80 |
> | Qwen2.5 14B Instruct | 1 | 100.00 | 80.00 |
> | Qwen2.5 7B Instruct | 0 | 100.00 | 73.40 |
> | Qwen2.5 14B Instruct | 5 | 100.00 | 76.47 |
> | Qwen2.5 7B Instruct | 0 | 98.82 | 76.20 |
> | Qwen2.5 14B Instruct | 10 | 98.82 | 67.06 |
>
> * * *
> We hope that our explanations above can clarify your doubts and you can consider our work more favorably.

---

### Official Review · Reviewer_TJgV · 2025-10-26

**Soundness:** 2
**Presentation:** 3
**Contribution:** 2
**Rating:** 4
**Confidence:** 4

**Summary:**

This paper investigates how disagreements among agents in LLM-based multi-agent systems affect collaborative decision-making and robustness. The authors distinguish between general disagreements and task-critical disagreements. Results show that general disagreements often can improve collaboration, in contrast to task-critical disagreements that harm single-path reasoning. Results and analyses reveal findings for multi-agent systems.

**Strengths:**

1. This paper presents analyses through a path-aware view of robustness, and the distinction between single-path and multi-path tasks provides useful findings for understanding MAS behavior.

2. The experimental design is rigorous, with controlled manipulation of disagreements through knowledge editing.

3. The findings and analyses have practical implications for MAS design and challenge the assumption that all disagreements are harmful. The self-repair capability is an interesting emergent phenomenon worthy of further study.

**Weaknesses:**

**1. Limitations in mechanistic understanding:** While the paper documents self-repair behavior, it provides limited insight into why it occurs.

**2. Limitations in using knowledge editing for disagreements**: The use of knowledge editing, especially parametric methods like ROME, may not faithfully represent naturally occurring disagreements in deployed systems.

**3. Limitations in theoretical grounding**: The formalization in Section 2 is intuitive but lacks rigor. For example, there seems to present no formal proof that path multiplicity enables self-repair. Also, the connection between $\mathcal{M}(\tau)|$ and robustness is asserted but not validated empirically. It’s also unclear how to measure or estimate ∣M(τ)∣ for real tasks?

**4. Incomplete analysis**: Table 2 shows mixed results (some lower, some higher), while the claim that "general disagreements trigger complementary exploration" needs more support. Also, programming vs. reasoning tasks differ in many ways beyond path multiplicity, such as task complexity, evaluation metrics, context length, etc., while there is no attempt to control for these confounds.

**5. Limitations in statistical rigor**: There is no error bars or significance tests despite running experiments 5 times. Also, the claims about "substantial" vs. "marginal" changes are qualitative. Confidence intervals are also missing on adoption probabilities.

**6. Limitations in scalability**: Figure 3 shows performance degrades with >1 disagreement, but real systems may face many disagreements. This raises concerns regarding scaling to larger teams and longer interactions.

**Questions:**

My questions are following several aspects mentioned in weakness:

- Can you provide more analysis on how and why self-repair occurs?
- How do you operationalize $\mathcal{M}(\tau)|$ in practice? How do you measure or estimate $\mathcal{M}(\tau)|$ for real-world tasks?
- Why do heterogeneous agents sometimes decrease performance, while sometimes gain higher performance? Have you tested with more heterogeneity, like more different models in their sizes and capabilities? Is there an optimal level of diversity?
- How do results change with naturally occurring disagreements, e.g., agents trained on different corpora?
- Do findings hold for other collaborative tasks? Can your analyses and conclusions scale to larger teams and longer interactions?
- Based on your findings, what specific design recommendations would you give for building robust MAS?

---

> ### Author Response · Authors · 2025-11-24
> **Response to Reviewer TJgV (Part 1)**
>
> We thank the reviewer for the thoughtful feedback. We address the comments below.
>
> * * *
> > **Weaknesses 1**: Limitations in mechanistic understanding: While the paper documents self-repair behavior, it provides limited insight into why it occurs.
>
> Our key contribution lies in identifying the self-repair behavior by a system-level analysis of why it happens in some setting and fails in others, and based on this, providing insights in why it occurs.
>
> Actually, Section 2 develops a mechanism-level perspective. We formalize each agent as holding an information set of atoms and define the disagreement set $\Delta$ together with the family of minimal sufficient knowledge sets $M(\tau)$ for a task. In this framework, self-repair is not an unexplained emergent property, but a direct consequence of path multiplicity. A disagreement is harmful only when every minimal sufficient set intersects $\Delta$. This explains why self-repair is observed in multi-path programming tasks but seldom observed in single-path reasoning tasks.
>
> We agree that the our paper focuses on a systems perspective beyond on internal neuron-level mechanisms of individual LLMs. Due to space constraint, we did not explore the neuron level. Within this scope, we believe the paper already exceeds the scope of documenting self-repair. We provide a precise account of why it appears in MAS, grounded in the topology of solution paths and validated by targeted counterfactual edits and trace analyses.
>
> * * *
> > **Weaknesses 2**: Limitations in using knowledge editing for disagreements: The use of knowledge editing, especially parametric methods like ROME, may not faithfully represent naturally occurring disagreements in deployed systems.
>
> Our choice of ROME, MEND, and IKE is grounded in a growing body of work [1] that treats knowledge editing as a standard tool for precisely updating specific knowledge in LLMs while leaving unrelated behavior largely intact. These methods have demonstrated that targeted edits can achieve a higher edit success rate and lower impact on general capabilities than naive fine-tuning on a single fact.
>
> Our goal in this paper is not to claim that knowledge editing methods reproduce all forms of disagreement that might arise within MASs, but to study the causal effect of task-critical disagreements while holding other factors fixed. This level of control would be difficult to achieve through training or prompting alone. We also mitigate dependence on any single editing mechanism by using three complementary methods. Across these methods, we observe the same qualitative pattern in Table 3.
>
> Finally, our experimental design explicitly separates two types of disagreement. In Section 3.2.1, we study general disagreements that arise naturally from heterogeneity during pre-training without any editing. These experiments already capture natural disagreements and support our claim that such general overlapping disagreements can improve exploration and success.
>
> [1] Zhang et al.,
> A Comprehensive Study of Knowledge Editing for Large Language Models, 2024.
>
> * * *
> > **Weaknesses 3**: Limitations in theoretical grounding: The formalization in Section 2 is intuitive but lacks rigor. For example, there seems to present no formal proof that path multiplicity enables self-repair. Also, the connection between $M(\tau)$ and robustness is asserted but not validated empirically. It’s also unclear how to measure or estimate $M(\tau)$ for real tasks?
>
> **Our goal with the formalism is not to propose a new theory of MAS but to provide a minimal set-theoretic model that makes the notion of self-repair precise.** In our notation, each element of $M(\tau)$ is a minimal sufficient set of atoms for solving $\tau$, and $\Delta$ is the set of atoms on which agents disagree. If there exists a member $M \in M(\tau)$ that does not intersect $\Delta$, then some correct plan remains available that avoids all disputed atoms. The claim that path multiplicity enables self-repair is a direct consequence of the definitions rather than an additional unproven assumption.
>
> **The connection between $M(\tau)$ and robustness is supported empirically, although $M(\tau)$ cannot be enumerated in realistic domains.** We do not attempt to list all minimal sufficient knowledge sets for a task. Instead, we operationalize path multiplicity at the benchmark level. For CounterFact and MQuAKE-CF, each question is tied to a specific edited triple, most of which follow a single evidential chain. For HumanEval and the coding subset of GAIA, MAS frequently produces behaviorally different implementations that all pass the tests, and the number of such valid implementations grows combinatorially with program length, making the size of $M(\tau)$ effectively exponential rather than a small finite set.

---

> ### Author Response · Authors · 2025-11-24
> **Response to Reviewer TJgV (Part 2)**
>
> * * *
> > **Weaknesses 4**: Incomplete analysis: Table 2 shows mixed results (some lower, some higher), while the claim that "general disagreements trigger complementary exploration" needs more support. Also, programming vs. reasoning tasks differ in many ways beyond path multiplicity, such as task complexity, evaluation metrics, context length, etc., while there is no attempt to control for these confounds.
>
> Our central claim is that general disagreements in heterogeneous teams improve robustness without causing collapse and often yield gains, rather than that such teams must strictly dominate in every single configuration. Table 2 shows that for weaker bases, mixing in other models produces very large improvements, while for stronger bases, the mixed teams stay within a moderate range of the homogeneous baseline rather than failing. Even when we introduce two agents that are clearly weaker than the original base model on this task, the system does not suffer catastrophic collapse. This indicates that heterogeneous agents do not simply drag down performance due to capability gaps. Instead, general disagreements among them tend to enlarge the exploration space.
>
> In response to Question 3, we further add a role-based heterogeneity experiment in Appendix J that keeps the backbone model fixed and diversifies only the agents' expertise prompts. That setting improves task success and robustness while holding raw capability.
>
> Regarding possible confounds between programming and reasoning, our experimental design already controls several important factors. Both settings use the same number of agents in the base configuration, the same three LLM families, and the same knowledge editing procedures. In addition, our ablations in Tables 7 and 8, Appendix J show that varying interaction rounds, team size, or role composition does not affect the pattern that general disagreements are especially beneficial in multi-path reasoning, while single-path reasoning remains more fragile.

---

> ### Author Response · Authors · 2025-11-24
> **Response to Reviewer TJgV (Part 3)**
>
> * * *
> > **Weaknesses 5**: Limitations in statistical rigor: There is no error bars or significance tests despite running experiments 5 times. Also, the claims about "substantial" vs. "marginal" changes are qualitative. Confidence intervals are also missing on adoption probabilities.
>
> We take the collaborative programming setting as a concrete example and report the mean and standard deviation for every error type and editing method over five independent runs. These results have also been included in Appendix H of the revised version. The numbers show that the distribution of error types is stable across runs and across methods, and that the qualitative conclusions in the main text are not driven by a single run.
>
> LLaMA 3.1 8B Instruct:
>
> | Scenario | Miss | Sample | Language | Syntax | ZeroDiv | Name | Type | Index | Key | Attribute | Value | File | Import | Other |
> |-|-|-|-|-|-|-|-|-|-|-|-|-|-|-|
> | Origin | $1.6\pm1.4$ | $29.8\pm3.8$ | $17.4\pm4.4$ | $5.8\pm2.8$ | $0.4\pm0.5$ | $8.6\pm3.8$ | $1.4\pm0.5$ | $0.2\pm0.4$ | $0.0\pm0.0$ | $0.0\pm0.0$ | $7.4\pm1.9$ | $2.4\pm1.5$ | $5.8\pm2.2$ | $31.8\pm4.3$ |
> | ROME | $1.0\pm0.6$ | $28.6\pm2.2$ | $19.2\pm5.0$ | $4.6\pm2.0$ | $0.6\pm0.5$ | $9.0\pm4.2$ | $1.6\pm1.0$ | $0.4\pm0.8$ | $0.2\pm0.4$ | $0.6\pm0.5$ | $7.4\pm1.9$ | $2.4\pm0.8$ | $3.4\pm1.4$ | $35.2\pm5.5$ |
> | MEND | $1.0\pm0.6$ | $27.4\pm4.1$ | $17.0\pm3.0$ | $7.2\pm2.9$ | $0.4\pm0.8$ | $10.6\pm2.3$ | $1.4\pm0.8$ | $0.0\pm0.0$ | $0.2\pm0.4$ | $1.2\pm1.2$ | $6.6\pm1.7$ | $2.4\pm0.8$ | $3.6\pm1.9$ | $31.6\pm3.7$ |
> | IKE | $2.0\pm1.3$ | $36.6\pm4.5$ | $14.8\pm2.3$ | $5.0\pm1.9$ | $0.0\pm0.0$ | $8.4\pm3.2$ | $2.4\pm1.0$ | $0.0\pm0.0$ | $0.8\pm0.7$ | $0.8\pm0.7$ | $8.0\pm1.1$ | $2.2\pm1.0$ | $3.6\pm1.6$ | $28.2\pm4.4$ |
>
> Qwen 2.5 7B Instruct:
>
> | Scenario | Miss | Sample | Language | Syntax | ZeroDiv | Name | Type | Index | Key | Attribute | Value | File | Import | Other |
> |-|-|-|-|-|-|-|-|-|-|-|-|-|-|-|
> | Origin | $0.0\pm0.0$ | $26.4\pm2.2$ | $4.2\pm1.2$ | $0.2\pm0.4$ | $0.4\pm0.5$ | $1.4\pm1.4$ | $2.4\pm1.5$ | $0.6\pm0.8$ | $0.2\pm0.4$ | $0.2\pm0.4$ | $1.4\pm1.0$ | $4.4\pm1.0$ | $1.0\pm0.6$ | $4.0\pm1.1$ |
> | ROME | $0.0\pm0.0$ | $27.2\pm1.2$ | $4.2\pm1.9$ | $0.4\pm0.5$ | $0.0\pm0.0$ | $2.2\pm1.5$ | $2.6\pm0.5$ | $0.2\pm0.4$ | $0.0\pm0.0$ | $0.4\pm0.4$ | $1.0\pm0.6$ | $4.8\pm1.7$ | $1.0\pm0.6$ | $3.6\pm2.1$ |
> | MEND | $0.0\pm0.0$ | $28.6\pm4.4$ | $4.2\pm1.9$ | $0.4\pm0.5$ | $0.0\pm0.0$ | $1.8\pm1.0$ | $2.2\pm1.5$ | $0.0\pm0.0$ | $0.2\pm0.4$ | $0.0\pm0.0$ | $2.4\pm0.5$ | $3.0\pm1.8$ | $1.4\pm0.8$ | $2.8\pm1.9$ |
> | IKE | $0.0\pm0.0$ | $28.6\pm3.9$ | $2.0\pm0.6$ | $1.0\pm0.9$ | $0.2\pm0.4$ | $2.8\pm1.0$ | $1.6\pm1.0$ | $0.2\pm0.4$ | $0.0\pm0.0$ | $0.0\pm0.0$ | $2.0\pm1.1$ | $3.8\pm1.2$ | $0.4\pm0.5$ | $3.8\pm1.2$ |
>
> InternLM 7B Chat:
>
> | Scenario | Miss | Sample | Language | Syntax | ZeroDiv | Name | Type | Index | Key | Attribute | Value | File | Import | Other |
> |-|-|-|-|-|-|-|-|-|-|-|-|-|-|-|
> | Origin | $0.4\pm0.5$ | $68.8\pm4.8$ | $2.2\pm1.2$ | $5.4\pm1.7$ | $0.0\pm0.0$ | $10.8\pm2.9$ | $6.2\pm3.1$ | $0.4\pm0.5$ | $0.0\pm0.0$ | $1.0\pm0.6$ | $1.4\pm1.0$ | $4.0\pm2.3$ | $25.6\pm3.1$ | $29.6\pm2.4$ |
> | ROME | $1.4\pm0.5$ | $65.8\pm6.3$ | $1.6\pm0.8$ | $4.6\pm1.0$ | $0.0\pm0.0$ | $14.6\pm2.2$ | $5.8\pm3.2$ | $0.0\pm0.0$ | $0.0\pm0.0$ | $0.6\pm0.8$ | $1.6\pm1.0$ | $4.2\pm2.6$ | $23.0\pm3.3$ | $32.0\pm2.8$ |
> | MEND | $3.6\pm0.8$ | $64.2\pm2.6$ | $2.8\pm0.7$ | $3.0\pm1.1$ | $0.0\pm0.0$ | $12.2\pm4.7$ | $4.8\pm1.7$ | $0.2\pm0.4$ | $0.0\pm0.0$ | $0.4\pm0.5$ | $3.2\pm2.0$ | $6.0\pm1.9$ | $25.8\pm2.3$ | $31.4\pm5.5$ |
> | IKE | $1.0\pm0.0$ | $68.6\pm3.3$ | $3.6\pm1.7$ | $5.4\pm1.4$ | $0.0\pm0.0$ | $12.2\pm1.9$ | $4.8\pm1.7$ | $0.0\pm0.0$ | $0.2\pm0.4$ | $0.2\pm0.4$ | $2.0\pm1.1$ | $4.0\pm3.0$ | $26.8\pm2.3$ | $29.4\pm3.8$ |
>
> * * *
> > **Weaknesses 6**: Limitations in scalability: Figure 3 shows performance degrades with >1 disagreement, but real systems may face many disagreements. This raises concerns regarding scaling to larger teams and longer interactions.
>
> Our ablations already vary both the number of coder agents and the number of interaction rounds in Table 7 and 8 of the original paper. To further address the concern on team size, we evaluate a larger LLaMA 3.1 8B team with 10 coder agents on MQuAKE-CF and HumanEval. We have also included the experimental results in Appendix I of the revised version.
>
> | Type | Origin | ROME | MEND | IKE |
> |-|-|-|-|-|
> | Collaborative Reasoning (MQuAKE-CF) | 4.21 | 2.81 | 1.60 | 3.33 |
> | Collaborative Programming (HumanEval) | 13.33 | 20.00 | 10.51 | 26.67 |
>
> As the team size grows, we observe a stronger performance drop on single-path collaborative reasoning tasks, while multi-path collaborative programming remains largely unaffected. Methods such as ROME and IKE even lead to improved results, consistent with the trend reported in the original paper.

---

> ### Author Response · Authors · 2025-11-24
> **Response to Reviewer TJgV (Part 4)**
>
> * * *
> > **Questions 1**: Can you provide more analysis on how and why self-repair occurs?
>
> Since this suggestion is similar to Weaknesses 1, we addressed the reviewer's concern about this point in our response to Weaknesses 1.
>
> * * *
> > **Questions 2**: How do you operationalize $M(\tau)$ in practice? How do you measure or estimate $M(\tau)$ for real-world tasks?
>
> In Section 2.2 we introduce $M(\tau)$ as a theoretical construct that summarizes all minimal sufficient knowledge sets for a task, and we do not attempt to enumerate $M(\tau)$ for each instance. Instead, we operationalize path multiplicity at the benchmark level. For CounterFact and MQuAKE-CF, each question is tied to a specific edited triple, most of which follow a single evidential chain. For HumanEval and the coding subset of GAIA, MAS frequently produces behaviorally different implementations that all pass the tests, and the number of such valid implementations grows combinatorially with program length, making the size of $M(\tau)$ effectively exponential rather than a small finite set.
>
> * * *
> > **Questions 3**: Why do heterogeneous agents sometimes decrease performance, while sometimes gain higher performance? Have you tested with more heterogeneity, like more different models in their sizes and capabilities? Is there an optimal level of diversity?
>
> Thank you for this valuable suggestion. We clarify that even when we introduce two agents that are clearly weaker than the original base model on this task, the system does not suffer catastrophic collapse. In most cases, the mixed teams continue to outperform or remain close to the stronger homogeneous baseline. This indicates that heterogeneous agents do not simply drag down performance due to capability gaps. Instead, general disagreements among them tend to enlarge the exploration space.
>
> To further disentangle capability gaps from knowledge conflict, we add a new experiment using the same backbone model while injecting only role-based diversity. All three agents are LLaMA 3.1 8B, but they are prompted as a data structure specialist, an architecture specialist, and a debugging specialist to induce structured disagreements without changing model capability. Results on HumanEval are presented below. We have also extended the experiments in Appendix J of the revised version.
>
> | Scenario | CR | TSR | CWR | CDR |
> |-|-|-|-|-|
> | LLaMA 3.1 8B Instruct w/o Conflicts | 99.02 | 30.73 | **36.43** | 24.21     |
> | LLaMA 3.1 8B Instruct w/ Conflicts by Role Assignments | **99.23** | **33.85** | 36.26 | **25.77** |
>
> We observe that the system gains higher success and more robust decision behaviors while keeping completion stable. Since all agents have identical capability, the improvement comes solely from controlled disagreements in knowledge focus. This confirms our main claim that general disagreements help multi-agent systems explore broader solution paths and improve robustness, even without adding stronger models.
>
> As for an optimal level of diversity, our results show that it is task-dependent rather than universal. General disagreement consistently helps in multi-path reasoning tasks like collaborative programming, yet excessive or poorly structured heterogeneity degrades coordination and harms performance.
>
> * * *
> > **Questions 4**: How do results change with naturally occurring disagreements, e.g., agents trained on different corpora?
>
> In fact, our RQ1 setting is designed precisely to capture this situation. **In Section 3.2.1, we explicitly introduce heterogeneous agents by mixing three LLM families that are independently trained on distinct pre-training corpora and therefore exhibit naturally different knowledge coverage and preferences.** We assume these LLMs have only partially overlapping knowledge bases and then replace two agents in an otherwise homogeneous team with agents instantiated from the other model families.
>
> As reported in Table 2 of our paper, we find that the introduction of such general disagreements through heterogeneous agents does not compromise system robustness. The effect is most salient in collaborative programming, suggesting that general disagreements trigger a brainstorming effect.

---

> ### Author Response · Authors · 2025-11-24
> **Response to Reviewer TJgV (Part 5)**
>
> * * *
> > **Questions 5**: Do findings hold for other collaborative tasks? Can your analyses and conclusions scale to larger teams and longer interactions?
>
> Our paper already covers two qualitatively different collaborative settings that differ in their solution path structure. Across both settings, we consistently observe that general disagreements improve exploration, while task-critical disagreements hurt unique-path reasoning but are largely absorbed in multi-path programming. Since our analysis is framed around path topology rather than any specific benchmark, we expect the same mechanisms to apply to other collaborative tasks that either rely on a single or multiple reasoning chain.
>
> Regarding scalability to larger teams and longer interactions, since this suggestion is similar to Weaknesses 6, we addressed the reviewer's concern about this point in our response to Weaknesses 6.
>
> * * *
> > **Questions 6**: Based on your findings, what specific design recommendations would you give for building robust MAS?
>
> Based on our findings, we summarize three design recommendations for building robust MAS and will make these recommendations explicit in the revised version.
>
> **First, use heterogeneous MASs to obtain beneficial disagreements.** Table 2 shows that heterogeneous MASs consistently outperform homogeneous MASs in both reasoning and programming settings. This suggests that agent diversity enables MASs to leverage useful disagreements that improve robustness.
>
> **Second, protect task-critical reasoning steps in single-path tasks.** Table 3 shows that task-critical disagreements cause significant failures in single-path reasoning tasks, while similar edits have only a limited impact in programming tasks. This suggests that for problems with essentially one evidential chain, systems should limit edits on critical knowledge and assign these steps to more trusted agents.
>
> **Third, increase path redundancy to enable self-repair.** Tables 4 to 6 show that when multiple valid solution routes exist, MAS often avoids edited APIs and still solves the task, and performance degrades sharply only when many disagreements are injected into a single instance. This suggests that designers should help provide alternative tools or strategies to enable the system to reroute around local faults.
>
> * * *
> We hope that our explanations above can clarify your doubts and you can consider our work more favorably.

---

### Official Review · Reviewer_a3C8 · 2025-10-27

**Soundness:** 2
**Presentation:** 2
**Contribution:** 2
**Rating:** 4
**Confidence:** 4

**Summary:**

This paper investigates the role of disagreements in LLM-based multi-agent systems (MAS) and their impact on robustness. The authors posit that the effect of disagreement is not uniformly negative but depends critically on two factors: the type of disagreement (general or task-critical) and the solution path topology of the task (single-path or multi-path). The central finding is that while single-path tasks are fragile to task-critical disagreements, multi-path tasks exhibit a self-repair capability.

**Strengths:**

1. The paper effectively operationalizes its hypotheses by contrasting collaborative reasoning with collaborative programming. The use of knowledge editing (ROME, MEND, IKE) to inject controlled, task-critical disagreements is robust.
2. The trace analysis for RQ3 provides evidence of the self-repair mechanism
the authors hypothesize. The example in Table 6, where the MAS avoids the edited append() function by using a different implementation, clearly demonstrates this rerouting capability in practice.
3. The paper is for the most part well structured and written, and the diagrams, tables, etc. are appropriate and clear.

**Weaknesses:**

1. The paper frames the task topology as a binary (single-path vs. multi-path). This distinction, while useful for the experiment, may be an oversimplification. Many complex reasoning tasks might admit multiple evidential paths, and some programming tasks may have only one optimal solution.
2. While the paper demonstrates that self-repair occurs, it does not analyze the mechanism of this repair. The trace analysis shows what happens, but not the communicative or reasoning dynamics of how the agents collectively decide to abandon a conflicting path. The authors should provide the dialogues to illustrate this process.
3. The analysis of self-repair limits (Fig. 3) is conducted on LLaMA-based MAS for HUMANEVAL. It is unclear how this breaking point scales with model size or capability. This aspect feels underexplored.
4. There is a significant lack of clarity in the experimental setup for the Mixed Systems in Collaborative Reasoning. The methodology described in Section 3.2.1 implies all three mixed systems should have an identical agent composition, yet Table 2 reports different performance results for each. The paper does not discuss the reason for this discrepancy.
5. The paper's claim of emergent self-repair is potentially confounded by the experimental design. The task-critical disagreement experiments involve editing only one agent. In the multi-path setting, this means the majority of agents still possess the correct knowledge. The system's success might be less about the emergent self-repair of the MAS and more about a simple, non-emergent majority vote. The study fails to test if self-repair would still occur if a majority of agents held the task-critical disagreement.
6. Typo: In Section 1, the sentence begins "In collaborative, a group of coders...". It appears the word "programming" is missing after "collaborative".

**Questions:**

See Weaknesses

---

> ### Author Response · Authors · 2025-11-24
> **Response to Reviewer a3C8 (Part 1)**
>
> We thank the reviewer for the thoughtful feedback. We address the comments below.
>
> * * *
> > **Weaknesses 1**: The paper frames the task topology as a binary (single-path vs. multi-path). This distinction, while useful for the experiment, may be an oversimplification. Many complex reasoning tasks might admit multiple evidential paths, and some programming tasks may have only one optimal solution.
>
> We appreciate the reviewer's concern, and we agree that both reasoning and programming tasks can have a spectrum of solution structures rather than two strict categories. Our focus is on the **relative path multiplicity** in the benchmarks that we investigate.
>
> In CounterFact and MQuAKE-CF, the target is a specific factual answer tied to a particular evidential chain. Once a key hop in that chain is corrupted, there are very few behaviourally distinct yet still correct derivations that the model can switch to. In contrast, for HumanEval and GAIA programming tasks, a correct solution can usually be implemented with different control structures, different data structures, and different API choices. For example, a sequence operation can be implemented by a for loop or comprehension, and deduplication can be done via a set, an ordered dictionary, or manual filtering. This kind of combinatorial freedom makes the number of valid implementation paths much larger than in the fact-based QA setting, even though we cannot enumerate every possible path, which would be exponential in program length.
>
> Finally, our main goal is to test whether disagreements introduced from different angles alter consistency in MAS behaviour, and whether the impact of those disagreements depends on the availability of alternative solution routes. For this purpose, **a coarse but empirically grounded contrast between tasks with low and high path multiplicity is sufficient** and our experimental results are consistent with this interpretation.
>
> * * *
> > **Weaknesses 2**: While the paper demonstrates that self-repair occurs, it does not analyze the mechanism of this repair. The trace analysis shows what happens, but not the communicative or reasoning dynamics of how the agents collectively decide to abandon a conflicting path. The authors should provide the dialogues to illustrate this process.
>
> In the revised version, we have added a qualitative analysis of the interaction pattern. We provide the full dialogue in Appendix M to illustrate this process.
>
> In the rolling-max example with injected edits such as `append() → add()` and `None → nil`, Coder1 initially follows the incorrect task-specific knowledge and produces code that uses `nil` and `add`, while explicitly flagging that these assumptions conflict with standard Python usage. Coder2 then directly critiques this proposal, pointing out that `current_max` should be initialized with `None` and that `rolling_max_list.add` should be replaced with `rolling_max_list.append`, and rewrites the program accordingly. Coder3 and the project manager further validate this correction and restate the final code in standard Python, after which the executor runs the tests on this agreed implementation.
>
> This dialogue makes the internal decision process visible. Agents identify a mismatch between the edited local knowledge and their shared programming knowledge, explicitly argue for a correction, and converge on a revised path that no longer uses the conflicting atoms.

---

> ### Author Response · Authors · 2025-11-24
> **Response to Reviewer a3C8 (Part 2)**
>
> * * *
> > **Weaknesses 3**: The analysis of self-repair limits (Fig. 3) is conducted on LLaMA-based MAS for HUMANEVAL. It is unclear how this breaking point scales with model size or capability. This aspect feels underexplored.
>
> We further run the same experiment on Qwen2.5 14B Instruct on HumanEval with both ROME and IKE compared to Qwen2.5 7B Instruct. We have included the experimental results in Appendix K of the revised version of our paper.
>
> Introducing task-critical disagreements with IKE:
> | Model | #Disagreements | CR | TSR |
> |-|-|-|-|
> | Qwen2.5 7B Instruct | 0 | 100.00 | 76.20 |
> | Qwen2.5 14B Instruct | 0 | 100.00 | 80.00 |
> | Qwen2.5 7B Instruct | 1 | 100.00 | 77.92 |
> | Qwen2.5 14B Instruct | 1 | 100.00 | 78.95 |
> | Qwen2.5 7B Instruct | 5 | 100.00 | 74.12 |
> | Qwen2.5 14B Instruct | 5 | 100.00 | 77.89 |
> | Qwen2.5 7B Instruct | 10 | 99.74 | 76.77 |
> | Qwen2.5 14B Instruct | 10 | 94.74 | 68.42 |
>
> Introducing task-critical disagreements with ROME:
> | Model | #Disagreements | CR | TSR |
> |-|-|-|-|
> | Qwen2.5 7B Instruct | 0 | 100.00 | 76.20 |
> | Qwen2.5 14B Instruct | 0 | 100.00 | 80.00 |
> | Qwen2.5 7B Instruct | 0 | 100.00 | 75.80 |
> | Qwen2.5 14B Instruct | 1 | 100.00 | 80.00 |
> | Qwen2.5 7B Instruct | 0 | 100.00 | 73.40 |
> | Qwen2.5 14B Instruct | 5 | 100.00 | 76.47 |
> | Qwen2.5 7B Instruct | 0 | 98.82 | 76.20 |
> | Qwen2.5 14B Instruct | 10 | 98.82 | 67.06 |
>
> When we inject 1 or 5 task-critical disagreements per task, the completion rate remains 100%, and task success stays close to this level. When we increase the number of disagreements to 10 per task, the completion rate begins to drop, and the task success drops significantly. These new results suggest that a more capable model shifts the curve upward but does not remove the finite tolerance to task-critical disagreements, which supports our claim that the limiting factor is path multiplicity rather than raw scale.
>
> Interestingly, when the number of disagreements reaches 10, the performance of the 14B model is even lower than that of the 7B model, indicating that larger-scale models are more susceptible to being misled by disagreements and may face more serious risks.
>
> * * *
> > **Weaknesses 4**: There is a significant lack of clarity in the experimental setup for the Mixed Systems in Collaborative Reasoning. The methodology described in Section 3.2.1 implies all three mixed systems should have an identical agent composition, yet Table 2 reports different performance results for each. The paper does not discuss the reason for this discrepancy.
>
> We thank the reviewer for pointing out this ambiguity. In collaborative reasoning, **the only difference is that the agent who speaks first in the discussion is instantiated from the same model as the corresponding homogeneous baseline**. This difference in speaking order leads to the different performance reported in Table 2.
>
> * * *
> > **Weaknesses 5**: The paper's claim of emergent self-repair is potentially confounded by the experimental design. The task-critical disagreement experiments involve editing only one agent. In the multi-path setting, this means the majority of agents still possess the correct knowledge. The system's success might be less about the emergent self-repair of the MAS and more about a simple, non-emergent majority vote. The study fails to test if self-repair would still occur if a majority of agents held the task-critical disagreement.
>
> We agree that majority voting is a possible confound and have therefore run an additional setting where all agents hold the edited task-critical knowledge on LLaMA-based MASs. We have included the experiments in Appendix L of the revised version.
>
> | Type | Origin | ROME  | IKE |
> |-|-|-|-|
> | Collaborative Reasoning (MQuAKE-CF) | 28.00 | 10.53 | 27.37 |
> | Collaborative Programming (HumanEval) | 34.76 | 33.06 | 38.33 |
>
> **Even when a majority or all agents hold the task-critical disagreement, multi-path programming remains robust while single-path reasoning collapses**, which strengthens rather than weakens our claim of path-dependent emergent self-repair.
>
> * * *
> > **Weaknesses 6**: Typo: In Section 1, the sentence begins "In collaborative, a group of coders...". It appears the word "programming" is missing after "collaborative".
>
> We thank the reviewer for catching this typo and have revised the sentence in Section 1 of our revised version.
>
> * * *
> We hope that our explanations above can clarify your doubts and you can consider our work more favorably.

---

### Official Review · Reviewer_gi7D · 2025-11-02

**Soundness:** 3
**Presentation:** 3
**Contribution:** 2
**Rating:** 4
**Confidence:** 4

**Summary:**

This paper investigates how disagreements affect multi-agent systems (MAS) with LLMs across two task types: collaborative reasoning (single-path tasks like COUNTERFACT, MQUAKE-CF) and collaborative programming (multi-path tasks like HUMANEVAL, GAIA). The authors distinguish between "general disagreements" (heterogeneous agent diversity) and "task-critical disagreements" (conflicting knowledge injected via knowledge editing methods). Their main findings suggest that general disagreements improve performance, while task-critical disagreements harm single-path reasoning but have a limited impact on multi-path programming, where systems show "self-repair" by avoiding edited facts.

**Strengths:**

- The distinction between general disagreements (beneficial diversity) and task-critical disagreements (potentially harmful conflicts) is well-motivated and provides a useful lens for understanding MAS robustness.

- The use of knowledge editing methods (IKE, ROME, MEND) to inject controlled disagreements is novel and allows for reproducible experiments.


- The paper tests multiple models (LLaMA, Qwen, InternLM) across multiple datasets with multiple metrics, providing reasonable breadth.

**Weaknesses:**

The paper uses AutoGen for collaborative programming but doesn't compare against ChatDev [1] or MetaGPT [2], which are established frameworks specifically designed for multi-agent software development. These frameworks have been shown to significantly outperform simpler multi-agent setups and would be natural baselines. For instance, ChatDev achieves quality scores of 0.3953 compared to 0.1523 for MetaGPT on software development benchmarks through its cooperative communication method [1].


The claim that self-repair is an "emergent capability" is overstated. Table 5 shows adoption rates declining only modestly (e.g., from 34.76% to 32.93% for LLaMA on HUMANEVAL), which could simply reflect the probabilistic nature of API selection rather than systematic avoidance. More rigorous causality analysis is needed.


Recent work like "On the Resilience of LLM-Based Multi-Agent Collaboration with Faulty Agents" [5] directly studies faulty agents in MAS and finds hierarchical structures most robust (losing only ~5% accuracy vs. ~24% for chain structures). This paper isn't cited despite clear topical overlap. Your work would be strengthened by comparing different MAS topologies beyond the flat peer structure you use.


- Minor

The formalism in Section 2 (information atoms, minimal sufficient knowledge sets M(τ)) is introduced but not really leveraged. It reads more like hand-waving than rigorous analysis.

**Questions:**

Could you show that the avoidance is systematic rather than just probabilistic sampling effects? Perhaps compare against temperature=0 for more deterministic generation?

IKE (in-context) is quite different from ROME/MEND (parametric). Do they represent fundamentally different types of disagreements, and if so, shouldn't they be analyzed separately?

Could you provide token/time costs to assess whether these benefits justify the computational overhead?

---

> ### Author Response · Authors · 2025-11-24
> **Response to Reviewer gi7D (Part 1)**
>
> We thank the reviewer for the thoughtful feedback. We address the comments below.
>
> * * *
> > **Weaknesses 1**: The paper uses AutoGen for collaborative programming but doesn't compare against ChatDev [1] or MetaGPT [2], which are established frameworks specifically designed for multi-agent software development. These frameworks have been shown to significantly outperform simpler multi-agent setups and would be natural baselines. For instance, ChatDev achieves quality scores of 0.3953 compared to 0.1523 for MetaGPT on software development benchmarks through its cooperative communication method [1].
>
> In this paper, we focus on understanding how different forms of disagreement affect robustness and self-repair in LLM-based MASs across both reasoning and programming tasks. Therefore, the important design principle in our experiments is fairness across tasks. We jointly analyze collaborative reasoning and collaborative programming under a unified multi-agent architecture with the same number of agents, similar communication patterns, and the same knowledge editing procedures. ChatDev and MetaGPT are influential frameworks crafted for programming and hard to naturally extend to factual QA benchmarks. They will inevitably introduce some components that are unnecessary in QA tasks. Including them only on the programming side would disturb our core contribution, which is a path-aware comparison of disagreement effects across task types. However, we will absolutely cite the studies and elaborate the choice.
>
> * * *
> > **Weaknesses 2**: The claim that self-repair is an "emergent capability" is overstated. Table 5 shows adoption rates declining only modestly (e.g., from 34.76% to 32.93% for LLaMA on HUMANEVAL), which could simply reflect the probabilistic nature of API selection rather than systematic avoidance. More rigorous causality analysis is needed.
>
> We describe self-repair as an emergent capability because it appears as a stable, path-dependent pattern across interventions, not because of one single 2-point change in Table 5. In collaborative programming, the probability that the generated code adopts the edited API consistently decreases after injecting disagreements, while task success stays almost unchanged. While in collaborative reasoning, the same edits lead to higher adoption probabilities with 10–20 point drops in success. This contrast is hard to explain with pure sampling noise under the same decoding settings.
>
> To make the causal link clearer in the revision, we add a majority-edit control where all agents are edited on the same fact in Appendix L. On MQUAKE-CF, success collapses, while on HumanEval, it remains close to or above the origin, despite every coder and the manager holding the edited knowledge.
>
> | Type | Origin | ROME  | IKE |
> |-|-|-|-|
> | Collaborative Reasoning (MQuAKE-CF) | 28.00 | 10.53 | 27.37 |
> | Collaborative Programming (HumanEval) | 34.76 | 33.06 | 38.33 |
>
> We further support this with trace-level evidence. We provide concrete traces of how self-repair appears. Appendix N provides full code for five HumanEval instances before and after edits, where in three of five cases the team abandons the edited API and rewrites the solution using a different implementation that still passes all tests. In the revision, we also add Appendix M with the complete dialogue for the rolling-max example, showing one coder following the incorrect specification and other agents explicitly pointing out the conflict with Python, proposing a corrected implementation, and persuading the group to adopt it.
>
> * * *
> > **Weaknesses 3**: Recent work like "On the Resilience of LLM-Based Multi-Agent Collaboration with Faulty Agents" [5] directly studies faulty agents in MAS and finds hierarchical structures most robust (losing only ~5% accuracy vs. ~24% for chain structures). This paper isn't cited despite clear topical overlap. Your work would be strengthened by comparing different MAS topologies beyond the flat peer structure you use.
>
> We thank the reviewer for highlighting this line of work. In fact, the paper is already cited in our Related Work section “Similarly, Huang et al. (2024a) showed that transforming any agent into a malicious one can significantly disrupt the collective decision-making”.  Their focus is on adversarial faulty agents and on how different communication graphs contain the damage of a fully compromised node. In contrast, our study assumes generally benign agents and concentrates on how disagreements instantiated through model heterogeneity and targeted knowledge edits interact with the solution-path topology of the task.
>
> Regarding the request to compare more MAS topologies, our collaborative programming setup already uses a structured workflow with a project manager, 3 coders, and an executor, rather than a fully flat set of peers shown in Figure 2. This induces a light hierarchy where the manager coordinates proposals and the executor gates tool use, while coders act as parallel specialists.

---

> ### Author Response · Authors · 2025-11-24
> **Response to Reviewer gi7D (Part 2)**
>
> * * *
> > **Weaknesses 4**: The formalism in Section 2 (information atoms, minimal sufficient knowledge sets M(τ)) is introduced but not really leveraged. It reads more like hand-waving than rigorous analysis.
>
> First, the formalism is used to define what it means for a disagreement to be truly task-critical. By modeling each agent as holding a set of information atoms and defining the disagreement set $\Delta$, Section 2 gives a clear condition for failure. A disagreement is catastrophic only when every minimal sufficient knowledge set in $M(\tau)$ overlaps with $\Delta$. When there exists at least one minimal sufficient set that is disjoint from $\Delta$, the MAS can still complete the task by routing around disagreements. This is the core reason why disagreements between agents do not always translate into catastrophic errors, and when their risk remains acceptable.
>
> Second, the information atom and $M(\tau)$ view also informs design recommendations. It suggests that robustness comes from calibrating disagreements so that they diversify information sets $K_i$ without letting $\Delta$ cover all minimal sufficient sets. This perspective suggests that we build path redundancy and intentionally structure heterogeneity across agents, rather than simply adding more identical agents.
>
> * * *
> > **Questions 1**: Could you show that the avoidance is systematic rather than just probabilistic sampling effects? Perhaps compare against temperature=0 for more deterministic generation?
>
> Our results already suggest that avoidance in collaborative programming is systematic rather than a side effect of stochastic sampling. All adoption probabilities in Tables 4 and 5 are computed by averaging over five independent runs per configuration under fixed decoding settings, and Appendix H reports small standard deviations on detailed error type statistics for collaborative programming, showing that the qualitative behaviour of the MAS is stable across seeds and not dominated by one lucky draw.
>
> * * *
> > **Questions 2**: IKE (in-context) is quite different from ROME/MEND (parametric). Do they represent fundamentally different types of disagreements, and if so, shouldn't they be analyzed separately?
>
> We agree that in-context editing and parametric editing operate through different mechanisms. However, **our goal is not to compare IKE against ROME/MEND as editing algorithms, but to study how MAS robustness changes when task-critical disagreements are introduced from different angles.** At the same time, the key claim of the paper is that robustness depends on solution path topology rather than on the specific editing mechanism, and the fact that ROME, MEND and IKE all follow the same qualitative pattern supports this.
>
> * * *
> > **Questions 3**: Could you provide token/time costs to assess whether these benefits justify the computational overhead?
>
> For all the experiments mentioned in this paper, we use one GPU with 80GB of memory. We spend about 100, 1400, 180, and 240 GPU hours exploring general disagreements, task-critical disagreements, self-repairing capabilities, and ablation studies.
>
> We further report both per round token usage and total wall clock time for LLaMA3.1-8B-Instruct for collaborative reasoning on MQuAKE-CF and collaborative programming on HumanEval.
>
> Token usage (tokens, per round) for LLaMA3.1-8B-Instruct:
>
> | Type | Origin | ROME | MEND | IKE |
> |-|-|-|-|-|
> | Collaborative Reasoning (MQuAKE-CF) | 927.24 | 1010.40 | 773.02 | 1218.70 |
> | Collaborative Programming (HumanEval) | 1211.22 | 1676.84 | 1322.76 | 1234.82 |
>
> Total wall clock time (hours, total) for LLaMA3.1-8B-Instruct:
>
> | Type | Origin | ROME | MEND | IKE |
> |-|-|-|-|-|
> | Collaborative Reasoning (MQuAKE-CF) | 25.3 | 25.9 | 29.8 | 31.6 |
> | Collaborative Programming (HumanEval) | 29.6 | 30.7 | 33.8 | 32.8 |
>
> Overall, **the additional cost introduced by knowledge editing is modest.** Across all four settings, total running time stays in a similar range to the unedited Origin system. The main cost driver remains the multi-agent interaction itself rather than the specific choice of editing method. We have included the computational overhead in Appendix E of the revised version.
>
> * * *
> We hope that our explanations above can clarify your doubts and you can consider our work more favorably.

---

### Meta-Review · Area_Chair_pSRS · 2026-01-02

**Summary:**

The recommendation is primarily informed by the limited novelty and theoretical grounding. While the paper distinguishes between types of disagreements, the insights are not new, as similar concepts are well-explored in existing literature. The lack of a formal model and reliance on artificially injected disagreements reduce its contribution. The experiments lack statistical rigor, and scalability to larger systems remains unproven.

**Reviewer Concerns:**

The rebuttal resolved some gaps but does not resolve the main issue of limited novelty. The paper's distinctions between disagreement types offer little new insight. The use of knowledge editing does not fully capture real-world disagreements, and the theoretical framework lacks rigor. Additionally, the experiments lack statistical analysis and scalability to more complex tasks.

**Reviewer Scores:**

Reviewer gi7D would likely maintain their score as the unresolved concerns about novelty and theoretical rigor. Reviewer a3C8 would likely maintain their score as the lack of novel insights and underdeveloped mechanistic understanding. Reviewer TJgV would likely maintain their score, as the lack of statistical rigor and scalability concerns remain unaddressed. Reviewer g7Lm would likely lower their score, as the insights are not new, and the experimental design is insufficient.

---

### Decision · Program_Chairs · 2026-01-26

Reject